# Modeling Asynchronous Time Series with Large Language Models

## Abstract

We present a novel prompt design for Large Language Models (LLMs) tailored to **Asynchronous Time Series**. Unlike regular time series, which assume values at evenly spaced time points, asynchronous time series consist of events occurring at irregular intervals, each described in natural language. Our approach effectively utilizes the rich natural language of event descriptions, allowing LLMs to benefit from their broad world knowledge for reasoning across different domains and tasks. This allows us to extend the scope of asynchronous time series analysis beyond forecasting to include tasks like anomaly detection and data imputation.

We further introduce **Stochastic Soft Prompting**, a novel prompt-tuning mechanism that significantly improves model performance, outperforming existing fine-tuning methods such as QLORA. Through extensive experiments on real-world datasets, we demonstrate that our approach achieves state-of-the-art performance across different tasks and datasets.

## 1 Introduction

An asynchronous time series (also named *temporal event sequence* or *continuous-time event sequence*) is a temporally ordered set of events that describe the progression of actions or occurrences. Asynchronous time series are ubiquitous in daily life, such as healthcare (Lorch et al., 2018; Rizoiu et al., 2018), finance (Bacry et al., 2015; Jin et al., 2020), e-commerce (Hernandez et al., 2017), and social media (Zhang et al., 2022; Kong et al., 2023). In each of those domains, predicting the next events plays a crucial role.

Unlike regular time series, which consist of values at evenly spaced time intervals (like weather measurements), asynchronous time series consist of multiple types of discrete events occurring sporadically over time. For example, in the context of social media platforms like Twitter, user interactions (likes, comments, shares, and follows) happen sporadically and at irregular intervals.(Zhao et al., 2015) Each such type of interaction with a user's profile represents an event type, and together with their timestamps, form an asynchronous time series. (Xue et al., 2024) Modeling such asynchronous time series is challenging due to the irregular timing and the diversity of event types, which contrasts with the uniformity and regularity of traditional time series data.Schirmer et al. (2022) (Horn et al., 2020) (Zhang et al.)

Traditionally, to model asynchronous time series, events are grouped into a fixed, small number of categorical types. (Xue et al., 2024) Separate stochastic processes—such as Poisson processes or Hawkes processes—are then modeled for each event type to predict which event will occur next and when. (Mei et al., 2022) (Hawkes, 1971) However, this approach presents several significant drawbacks. *Firstly*, it inherently limits research to datasets with a small number of event types because modeling each event type separately becomes increasingly computationally intensive as the number of event types grows (Zuo et al., 2020). *Secondly*, events can vary widely and may not fit neatly into predefined categories. *Thirdly*, this method leads to the loss of meaningful natural language descriptions associated with the events. *Fourthly*, these methods treat each event type independently, ignoring any interactions between them - for example, likes and shares of a tweet are not independent events. *Lastly*, extending these methods to other tasks require significant theoretical development. (Shchur et al., 2021)

Deep learning models have significantly revolutionized techniques for time series modeling, and even more so with the introduction of transformers (Vaswani et al., 2017). However, there are

often limitations due to the scarcity of training data, overfitting in specific domains, and the highly specialized architectural designs. In response to those challenges, Large Language Models (LLMs) have emerged as a powerful and promising direction to model time series data. For example, (Gruver et al., 2023; Zhou et al., 2023; Xue & Salim, 2023; Jin et al., 2024) have illustrated how LLMs can be used as time series forecasters when the input time series is encoded as a string of numeric digits, by casting the time series forecasting problem as a next-token prediction in text, hence unlocking the use of powerful pre-trained models. LLMs have also been explored in other domains like action forecasting from videos (Zhao et al., 2024; Wang et al., 2024). However, these approaches focus on regular time series with evenly spaced numerical observations and cannot be directly applied to asynchronous time series due to their irregular intervals and diverse event types described in natural language. While LLMs have recently been explored for action recognition and action forecasting from videos Zhao et al. (2024) Wang et al. (2024), applying LLMs to textual asynchronous time series over multiple tasks (like anomaly detection and imputation) remains largely unexplored.

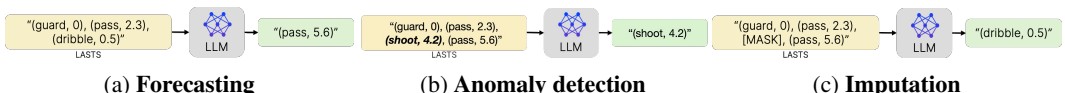

(a) **Forecasting**    (b) **Anomaly detection**    (c) **Imputation**

Figure 1: We show that our LASTS framework can solve following tasks on asynchronous time series data: (a)**Forecasting:** The model is given a sequence of events, encoded as text, with the goal of predicting the next event. (b)**Anomaly detection:** The model is given a sequence of events containing an incorrect event (bold) with the goal of finding the incorrect event. (c)**Imputation:** The model is given a sequence of events containing a masked event, encoded as text, with the goal of predicting the masked event.

This paper presents **LASTS** (**L**anguage-modeled-**As**ynchronous **T**ime **S**eries), a novel prompting based framework to adapt LLMs to asynchronous time series data while keeping the backbone model intact. To the best of our knowledge, this is the first work to explore the capabilities of LLMs to process textual asynchronous time series data and works on multiple tasks as shown in Figure 1. Our framework overcomes the drawbacks presented by traditional approaches for modeling asynchronous time series- It can handle datasets with large number of event types easily, it does not need to group events into predefined categorical bundles, it retains the natural language descriptions of event types and utilizes them, and it is able to leverage the rich interactions between different event types.

Our contributions can be summarized as follows:

- **We introduce LASTS (Language-modeled Asynchronous Time Series)** which is a novel framework that leverages Large Language Models (LLMs) to model asynchronous time series data, while effectively handling datasets with a large number of event types without the need for predefined categorical groupings. To the best of our knowledge, this is the first work to explore the capabilities of LLMs to process textual asynchronous time series data across multiple tasks such as forecasting, anomaly detection, and data imputation.

- **We introduce Stochastic Soft Prompting (StoP)** which is an innovative prompt-tuning mechanism that serves as a parameter-efficient method to adapt LLMs to asynchronous time series data. StoP learns soft prompts that significantly improve model performance and enhance adaptability by randomly truncating the prompts during training to learn more diverse representations.

- **We conduct comprehensive evaluations** on real-world datasets across multiple tasks to demonstrate the effectiveness of our proposed method. Our approach achieves state-of-the-art performance, outperforming existing methods, and highlights the potential of LLM-based models to effectively process and analyze asynchronous time series data.

## 2 RELATED WORK

**Temporal Point Processes (TPPs).** TPPs (Hawkes, 1971; Daley & Vere-Jones) have emerged as the standard method to model asynchronous time series data. Over the last decade, a large number of neural temporal point processes have been proposed to capture complex dynamics of stochastic

processes in time by using neural networks. Du et al. (2016); Mei & Eisner (2017) proposed to use models based on Recurrent Neural Networks (RNNs) to model the sequence of events. Then, more advanced models (Mehrasa et al., 2019; Lüdke et al., 2023) were proposed to better model the uncertainty when predicting the future. Recently, several neural TPP models incorporate Transformers in order to improve the performance by using attention to better model long-term dependencies: Self-attentive Hawkes process (SAHP) (Zhang et al., 2020), Transformer Hawkes process (THP) (Zuo et al., 2020), and Attentive Neural Hawkes Process (A-NHP) (Mei et al., 2022).

**Transformers for Time Series.** Transformers (Vaswani et al., 2017) have become popular to model regularly-sampled time series because of their ability to capture long-range dependencies and to extract semantic correlations among the elements of a long sequence. Informer (Zhou et al., 2021) introduced a novel self-attention architecture to reduce the quadratic complexity of the original self-attention. Autoformer (Wu et al., 2021) used a novel decomposition architecture with an auto-correlation mechanism to identify more reliable temporal patterns. Crossformer (Zhang & Yan, 2023) proposed a novel architecture to model both the cross-time and cross-dimension dependencies multivariate time series forecasting. PatchTST (Nie et al., 2023) tokenizes the time series in patches, and proposes a channel-independent patch time series Transformer to improve the long-term forecasting accuracy.

Due to the space limitations, we only review some popular models and invite the reader to check out (Wen et al., 2023; Zeng et al., 2023) for a more complete literature reviews of Transformer models for regularly-sampled time series. Most of the time series Transformer models are designed for specific tasks, and cannot be easily extended to asynchronous time series data or other tasks like anomaly detection or imputation.

**Foundation Models (FMs) for Time Series.** FMs (Bommasani et al., 2021) are a family of deep models that are pretrained on vast amounts of data, and have caused a paradigm shift due to their unprecedented capabilities for zero-shot and few-shot generalization. FMs have revolutionized natural language processing (Brown et al., 2020; BigScience Workshop et al., 2023; Wu et al., 2024; Dubey et al., 2024) and computer vision (Radford et al., 2021; Kirillov et al., 2023). The availability of large-scale time series datasets has opened the door to pretrain a large model on time series data. ForecastPFN (Dooley et al., 2024) proposed the first zero-shot forecasting method trained purely on synthetic data. Lag-Llama (Rasul et al., 2023) introduced a univariate probabilistic forecasting model that was pretrained on a large corpus of diverse time series data. TimeFM (Das et al., 2024) pretrained a decoder style attention model with input patching, using a large time series corpus comprising both real-world and synthetic datasets. Chronos (Ansari et al., 2024) introduced a framework for pretraining on tokenized time series data, achieving state-of-the-art zero-shot forecasting performance and simplifying forecasting workflows. MOIRAI (Woo et al., 2024) is an enhanced Transformer architecture pretrained in the Large-scale Open Time Series Archive, that achieves competitive performance as a zero-shot forecaster.

**LLMs for Time Series** LLMs pretrained on large amounts of text data have emerged as a promising direction to model time series data. GPT4TS (Zhou et al., 2023), LLM4TS (Chang et al., 2023), and TEMPO (Cao et al., 2023) fine-tuned a pretrained GPT2 (Radford et al., 2019) on some time series downstream tasks to capture intrinsic dynamic properties. TimeLLM (Jin et al., 2024) proposed a reprogramming framework to repurpose LLMs for general time series forecasting with the backbone language models kept intact. PromptCast (Xue & Salim, 2023) introduced a new prompt-based forecasting paradigm, where the numerical input and output are transformed into prompts and the forecasting task is framed in a sentence-to-sentence manner. LLMTime (Gruver et al., 2023) showed that LLMs can zero-shot extrapolate time series if the numerical values of the time series are well represented. LLM Processes (Requeima et al., 2024) explores various prompt configurations for using LLMs for time series forecasting condiitoned on a textual context. We refer the reader to (Zhang et al., 2024) for a more detailed survey on the topic.

**Vision Models for Time Series.** Several works started to explore the use of FMs pretrained on images because of the better intrinsic similarities between images and time series such as trend, stationarity, seasonality/periodicity, and sudden change. (Zhou et al., 2023) tried to fine-tune a BEiT (Bao et al., 2022) trained on images for time series forecasting, but it falls short of the leading text-based and time series-based FMs. Recently, VisionTS (Chen et al., 2024) proposes to use a vision

Transformer pretrained on ImageNet to reduce the cross-domain gap or in-domain heterogeneity between time series and text.

**Parameter Efficient Fine Tuning (PEFT).**    PEFT (Mangrulkar et al., 2022) is a paradigm to adapt pretrained LLMs to various domains without fine-tuning all of a model's parameters, which can be costly and require large amount of training data. LoRA (Hu et al., 2021) methods freeze the pretrained model weights and injects trainable rank decomposition matrices into each layer of the Transformer architecture, greatly reducing the number of trainable parameters for downstream tasks. QLoRA (Dettmers et al.) advances finetuning by significantly reduces memory usage while preserving task performance.

**Soft Prompt Tuning.**    Soft prompts have emerged as a compute efficient method for adapting a pretrained LLMs to new domains without altering their core architectures. Brown et al. (2020) was among the first to demonstrate the power of prompting for task adaption of pretrained language models, but automatically finding suitable sets of text prompts remains an open challenge. Li & Liang (2021); Qin & Eisner (2021) proposed the prefix tuning technique that preprends a few task specific soft tokens to the input and hidden states of each Transformer layer. During the training, the parameters of soft prompts are updated by gradient descent while the model parameters keep frozen. Liu et al. (2021) showed the prefix tuning technique could be effectively applied to natural language understanding with different scales of models. Lester et al. (2021) simplified the prefix tuning technique that only adds soft prompts to the input layer and is now considered as the standard soft prompt-tuning.

## 3   BACKGROUND

**Notations.**    We observe $n$ events over a fixed time interval $[0, T)$, with each event being denoted as $(e, t)$, where $e \in \mathcal{E}$ is the event type (or attributes) and $\mathcal{E}$ represents the space of event types. An *asynchronous time series* is a sequence of events $x_{1:n} = ((e_1, t_1), (e_2, t_2), \ldots, (e_n, t_n))$ where $t_i$ is an increasing sequence in $[0, T)$ that does not necessarily observe any periodicity. A common alternative to the event time $t_i$ is the inter-arrival time $\tau_j := t_j - t_{j-1}$, they are considered isomorphic and often used interchangeably. In our work there is very little constraint on $\mathcal{E}$ and in principle, our model still works even if $\mathcal{E}$ is infinite. We only need to be able to compute a vectorial representation of the event type/attributes, which is achieved through the LLM's learned input embeddings in our work.

**Language modeling.**    Language modeling is a widely used task to train LLMs where the goal is predicting the next word or character in a document. Language models are designed to work on a sequence of $m$ tokens, where each token belongs to a vocabulary. Tokenizer transforms the input text data into a sequence of tokens. The tokenization process is important and can impact significantly performances, for it directly influences how patterns form within tokenized sequences and the types of operations that language models can learn.

**Tasks**    We propose a new approach to model asynchronous time series with LLMs which solves three different tasks (see Figure 1):

- **Forecasting** (also known as *next event prediction*) Given a history of events $x_{1:m}$ from an asynchronous time series, the model is tasked with predicting the next event $x_{m+1}$.
- **Data imputation.** One of the events $x_j$ of the series is randomly chosen and masked, the model is tasked with filling in the gap.
- **Anomaly detection.** One event $x_j$ of the series is randomly chosen and its event type $e_j$ is replaced randomly by another event type $e'$. The model is tasked with identifying this out-of-place element.

To find the right recipe for the model to solve these tasks, we innovated in two major directions: first, we studied various representations of the asynchronous time series as inputs to LLMs (Section 4.1) for zero shot completion of these tasks; and secondly, we study different parameter efficient techniques to adapt an LLM backbone for working with asynchronous time series, while leveraging its knowledge of the world and its understanding of natural language (Section 4.2).

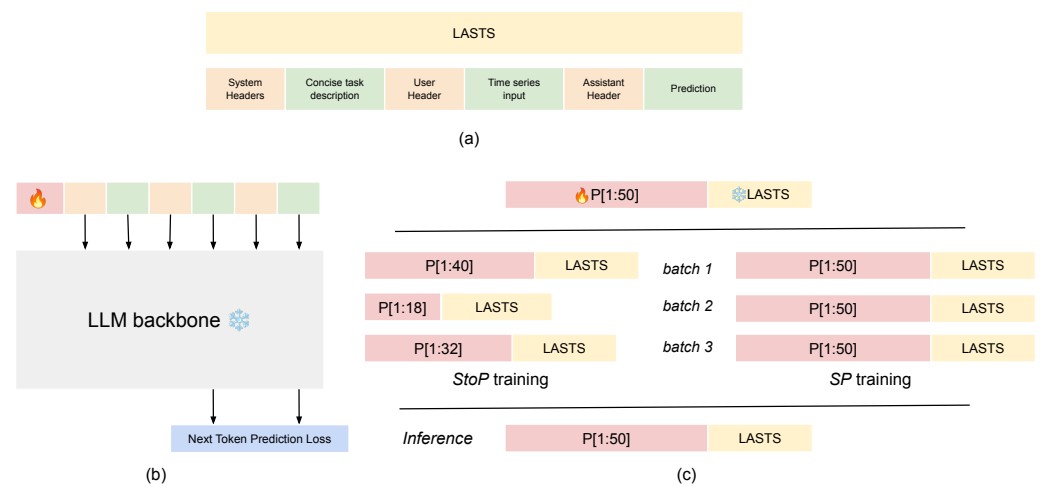

Figure 2: Overview of LASTS and Prompt Tuning with LASTS: (a) Structure of a LASTS prompt for adapting an asynchronous time series for use as input for an LLM. (b) $SP$ and $StoP$ training setup - the LLM backbone is frozen and the soft prompt is fine-tuned via gradients computed through the standard next token prediction loss. (c) Comparison of SP and StoP training. In StoP, a random prefix of the trainable prompt is used during each training batch.

## 4 PROPOSED METHOD

### 4.1 **LASTS** - PROMPTING LLMs WITH ASYNCHRONOUS TIME SERIES DATA

Unlike ordinary time series, often represented as sequences of numerical values (Gruver et al., 2023), asynchronous time series are represented as sequences of events $x_i = (e_i, t_i)$, where $e_i$ is the event type, and $t_i$ is a representation of the timestamp of this event. Normally, $t_i$ is expressed as inter-arrival time, which is the time elapsed between event $x_{i-1}$ and $x_i$.

In prior work on modeling asynchronous time series (Du et al., 2016; Mehrasa et al., 2019; Zhang et al., 2020; Mei et al., 2022), events are typically reduced to categories from a small set of options. In contrast, we retain the event types $e_i$ as natural language descriptions. We introduce LASTS, which specifies how to input an asynchronous time series as part of a prompt to effectively leverage LLMs for various tasks on such data.

**LASTS Prompt Structure** The LASTS prompt consists of three parts that can be mapped to the system-user-assistant structure when using an instruction fine-tuned LLM (see Figure 2). The **system prompt** introduces what an asynchronous time series is, provides a description of the task to be performed, and includes details about the underlying dataset. The **user prompt** represents the input series as a comma-separated sequence of tuples $(e_i, t_i)$, where $e_i$ is the textual description of the event type and $t_i$ is the inter-arrival time. The **assistant prompt** contains the correct event if performing LLM adaptation training, or is left to be generated by the LLM during inference.

More details about the exact prompts used in our experiments can be found in Appendix A.2.

### 4.2 PARAMETER EFFICIENT LLM ADAPTATION WITH LASTS REPRESENTATION

Having established a representation of asynchronous time series for use with LLMs via LASTS, we further enhance the model's adaptability to various tasks using three different adaptation techniques:

**Low Rank Adaption** LoRA is a family of low-rank adaptation techniques that reduce the number of trainable parameters by learning small, low-rank updates to selective model weights, allowing for efficient fine-tuning of large models. We adapt the LLM backbone for our tasks by applying low-rank adaptations using the LASTS representation as inputs to encode both the task and the input asynchronous time series.

**Soft Prompting (SP)**   SP involves prepending a continuous prompt to the LASTS representation, which is trained through gradients from next token prediction loss. This guides the model towards task-specific behavior without altering the model weights directly. (See Figure 2)

**Stochastic Soft Prompting (StoP)**   We propose Stochastic Soft Prompts - an enhancement of SP which learns more robust prompts by imposing a coarse-to-fine structure on the prompt tokens. (See section 5.3). Similar to SP, we prepend a continuous prompt to the LASTS representation which is trained through gradients from next-token prediction loss. However, in SP, the entire soft prompt $P$ of length $L$ is used during training, while in StoP, we randomly select a prefix of the prompt $P$ for each training batch. Specifically, for each batch, we choose a prefix length $l$ from a probability distribution $p(l)$, where $l \leq L$. The soft prompt used for that batch is then represented by:

$$P_{batch} = P[:l] \ \text{ with } \ l \sim p(l) \tag{1}$$

In our experiments, we use a uniform distribution as $p$. Both the forward pass and the backward pass are conducted using only the selected prefix $P_{\text{batch}}$. During inference, we use the entire learned soft prompt of length $L$:

$$P_{\text{inference}} = P[1:L] \tag{2}$$

See figure 2 for more details. Our approach is inspired by techniques like dropout (Srivastava et al., 2014) and stochastic depth (Huang et al., 2016), as well as audio models like SoundStream (Zeghidour et al., 2021), where randomly selecting the first $k$ codebooks during training enables better generalization.

These adaptation techniques enable an LLM backbone to handle a variety of asynchronous time series tasks, including forecasting, imputation, and anomaly detection, while maintaining parameter efficiency. Details on the exact prompt representation are provided in Appendix A.4.

## 5 EXPERIMENTS

### 5.1 EXPERIMENTAL SETUP

**Datasets.**   We perform experiments on two different sets of datasets: three text-based action datasets and five standard temporal point process datasets. The main difference is that actions are represented by words in the action datasets, whereas they are represented by indices in temporal point process datasets. The **text-based action datasets** are built from the action annotations of activity videos. *Breakfast* (Kuehne et al., 2014) contains 1712 videos with 177 action classes related to breakfast preparation. Each video has a sequence of events to prepare breakfast, with each event containing the timestamp and the action. *EPIC-KITCHENS-100* (Damen et al., 2022) is a large-scale dataset in egocentric vision capturing daily activities in the kitchen over multiple days with a total of 100 hours of recording. It presents more complex activity than Breakfast dataset, with rich annotations of sequences of actions comprising of 97 verb classes and 300 noun classes, with 20K unique narrations. *MultiTHUMOS* (Yeung et al., 2018) contains 400 videos with 65 action classes related to human activities. Each video has a sequence of human activity events, with each event containing the timestamp and the activity. For the **temporal point process datasets**, we use the five benchmarks introduced in (Xue et al., 2024): *Amazon* (Ni et al., 2019) where the goal is to predict the timestamp and category (among 16 categories) of the next reviewed product, *Retweet* (Zhou et al., 2013) where the goal is to predict the timestamp and category (among 3 categories) of the next user to retweet a post, *Taxi*[1] where the goal is to predict the timestamp and category (among 10 categories) of the next pick-up or drop-off of a taxi driver, *Taobao* (Xue et al., 2022) where the goal is to predict the timestamp and category (among 20 categories) of the item clicked by a user, and *StackOverflow*[2] where the goal is to predict the timestamp and category (among 22 categories) of the next badges for a given user. We follow the same data preprocessing as in (Xue et al., 2024). For each of these datasets, the semantic meaning of the event type is unknown, and only the index of the event type is available. We use the index of the event type as input of our model.

---

[1] https://chriswhong.com/open-data/foil_nyc_taxi/
[2] https://snap.stanford.edu/data/

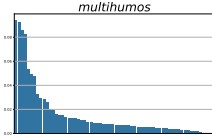 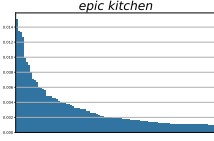 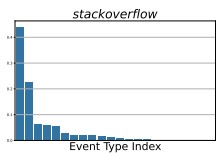

Figure 3: Visualization of normalized count (y-axis) w.r.t the event type sorted by count (x-axis) for four of the datasets. We observe these datasets are imbalanced. Using the macro-F1 gives the same importance to all the classes, whereas the accuracy gives more importance to majority classes.

**Metrics.** Due to bi-modality nature of the asynchronous time series, we report separate metrics for the event type and time. We report the Macro-F1 (M-F1) (Yang, 1999) for event type prediction as Macro-F1 is better suited for multi-class classification tasks with skewed class distributions (see Figure 3) than accuracy because Macro-F1 gives the same importance to all the classes. We report the Mean Absolute Error (MAE) or Root Mean Square Error (RMSE) for time prediction, as both metrics are used based on the dataset.

**Implementation details** We use Llama-3-8B-Instruct (Dubey et al., 2024) as our LLM backbone. For zero-shot experiments, we disable sampling during response generation, ensuring deterministic outputs. For LLM adaptation experiments, we use QLoRA as the low rank adaptation algorithm, Adam as the optimizer, and a constant learning rate of $2e^{-4}$ for QLoRA and $1e^{-4}$ for prompt tuning. Following Xue et al. (2024), we split our datasets into a train/validation/test ratio of 70/10/20. Both SP and StoP training are conducted for the same number of epochs. We employ early stopping based on the Macro-F1 on the validation set. We report performance on the test set.

We use a prompt length of $400$ for prompt tuning in both SP and StoP experiments. This value was selected through hyperparameter tuning across all datasets and tasks, striking a balance between model capacity, performance, and the compute resources available to us. Given that Llama-3-8B-Instruct has a hidden dimension of $4096$, this configuration results in approximately $1.6M$ trainable parameters, which corresponds to only $0.02\%$ of the LLM parameters. For QLoRA, we use a rank of 4, resulting in a comparable number of trainable parameters ($1.7M$)

## 5.2 EXPERIMENT RESULTS

**Baselines** We evaluate our methods using four sets of baselines. See Appendix A.5 for details.

- **Random baseline**: We establish a random baseline simulating random guesses to evaluate our methods on the three text-based datasets and tasks. (See Table 1)

- **Foundation models for time series**: We use a state-of-the-art pretrained foundation model for time series forecasting, **Chronos** Ansari et al. (2024), as a baseline for forecasting and imputation tasks on asynchronous time series. (See Table 1)

- **LLM for time series**: We adapt two LLM-based time series forecasting methods, **LLM-Time** (Gruver et al., 2023) and **LLM Processes** (Requeima et al., 2024), as baselines for zero-shot LASTS prompting on asynchronous time series. (See Table 1)

- **TPP models**: We compare our model with state-of-the-art TPP models for asynchronous time series (Xue et al., 2024). We report the results for two popular RNN-based models: Recurrent marked temporal point process (RMTPP) (Du et al., 2016) and neural Hawkes Process (NHP) (Mei & Eisner, 2017). We also compare with three attention-based models: self-attentive Hawkes process (SAHP) (Zhang et al., 2020), Transformer Hawkes process (THP) (Zuo et al., 2020), attentive neural Hawkes process (AttNHP) (Yang et al., 2022). (See Table 2)

**Results** Our results on the the three tasks (forecast, imputation, anomaly detection) and the three text datasets (Breakfast, MultiTHUMOS, EPIC-KITCHENS) are presented in Table 1. Based on our results, we make 5 main observations. Firstly, LASTS proves to be an effective and robust representation for asynchronous time series data across multiple datasets. LASTS Zero Shot consistently outperforms the Time Series Foundation Model Chronos and LLM-based methods (LLMTime and

| Model | Breakfast | | | MultiTHUMOS | | | EPIC-KITCHENS | | |
|---|---|---|---|---|---|---|---|---|---|
| | M-F1 ↑ | MAE ↓ | ACC ↑ | M-F1 ↑ | MAE ↓ | ACC ↑ | M-F1 ↑ | MAE ↓ | ACC ↑ |
| | Forecast | | | | | | | | |
| Random | 0.0162 | 40.1513 | 0.0201 | 0.0417 | 1.8803 | 0.0382 | 0.0000 | 3.2001 | 0.0001 |
| Chronos | 0.0011 | 43.0502 | 0.0021 | 0.0265 | 1.9805 | 0.0279 | 0.0000 | 3.5925 | 0.0005 |
| LLMTime | 0.0240 | 37.3902 | 0.0288 | 0.1280 | 2.2060 | 0.1235 | 0.0040 | 4.8948 | 0.0458 |
| LLM Processes | 0.0337 | 44.9856 | 0.0845 | 0.1278 | 2.0471 | 0.0970 | 0.0049 | 4.3843 | 0.0703 |
| LASTS Zero Shot* | 0.0604 | 38.1630 | 0.0969 | 0.1361 | 1.8868 | 0.1826 | 0.0105 | 3.1566 | 0.0920 |
| LASTS Few Shot* | 0.1518 | 35.5605 | 0.2133 | 0.1676 | 1.8114 | 0.2581 | 0.0149 | 3.3092 | 0.1150 |
| LASTS + QLORA* | 0.2558 | 33.9737 | 0.3763 | 0.3218 | 1.7281 | 0.4337 | 0.0764 | 2.8964 | 0.2160 |
| LASTS + SP* | 0.2341 | 32.8417 | 0.3691 | 0.3707 | 1.6630 | 0.4782 | 0.0780 | **2.8830** | 0.2217 |
| LASTS + StoP* | **0.2633** | **32.5464** | **0.3880** | **0.3947** | **1.6503** | **0.4784** | **0.0797** | 3.0318 | **0.2298** |
| | Imputation | | | | | | | | |
| Random | 0.0168 | 37.7029 | 0.0214 | 0.0435 | 2.3622 | 0.0416 | 0.0000 | 3.4269 | 0.0001 |
| Chronos | 0.0013 | 38.4039 | 0.0044 | 0.0294 | 2.3971 | 0.0312 | 0.0000 | 3.6955 | 0.0000 |
| LLMTime | 0.0137 | 35.9899 | 0.0381 | 0.0968 | 2.6998 | 0.1330 | 0.0005 | 3.6750 | 0.0314 |
| LLM Processes | 0.0156 | 34.7117 | 0.0488 | 0.1123 | 2.3786 | 0.1430 | 0.0008 | 4.2600 | 0.0371 |
| LASTS Zero Shot* | 0.0263 | 33.0097 | 0.0594 | 0.0915 | 2.6696 | 0.1210 | 0.0015 | 3.6527 | 0.0446 |
| LASTS Few Shot* | 0.0520 | 33.3440 | 0.1001 | 0.1013 | 2.3982 | 0.1569 | 0.0023 | 3.2528 | 0.0547 |
| LASTS + QLORA* | 0.1688 | 28.5638 | 0.2500 | 0.2132 | **2.2179** | 0.2744 | 0.0378 | 3.1194 | 0.1137 |
| LASTS + SP* | 0.1581 | 28.8503 | 0.2264 | 0.2044 | 2.4092 | 0.2528 | 0.0423 | 3.1456 | 0.1270 |
| LASTS + StoP* | **0.2064** | **28.2251** | **0.2740** | **0.2213** | 2.3445 | **0.2839** | **0.0610** | **3.1116** | **0.1424** |
| | Anomaly Detection | | | | | | | | |
| Random | 0.0349 | — | 0.0396 | 0.0381 | — | 0.0552 | 0.0238 | — | 0.0307 |
| LLMTime | 0.0240 | — | 0.0288 | 0.0415 | — | 0.0639 | 0.0048 | — | 0.0650 |
| LASTS Zero Shot* | 0.0923 | — | 0.0763 | 0.2755 | — | 0.1949 | 0.0159 | — | 0.0777 |
| LASTS Few Shot* | 0.0837 | — | 0.0563 | 0.3535 | — | 0.2720 | 0.0337 | — | 0.1637 |
| LASTS + QLORA* | 0.7011 | — | 0.6478 | 0.6003 | — | 0.5084 | 0.6520 | — | 0.6988 |
| LASTS + SP* | 0.6520 | — | 0.5937 | 0.5231 | — | 0.4657 | 0.6159 | — | 0.6635 |
| LASTS + StoP* | **0.7198** | — | **0.6698** | **0.6045** | — | **0.5168** | **0.6603** | — | **0.7037** |

Table 1: Performance of our models on three textual datasets for forecasting, imputation, and anomaly detection tasks. Metrics are macro F1, and accuracy (ACC) for event type prediction and MAE for event time prediction. The **best result** in each class is highlighted in bold, and the second-best result is underlined. Note that for anomaly detection, since the task involves identifying only the anomalous event, the MAE metric is not applicable and Chronos and LLM Processes are not adaptable (see A.5). A * indicates our method. We use 5 examples for few shot results (see A.9).

LLM Processes) in most evaluations, highlighting the advantage of using textual event descriptions enabled by LASTS. Secondly, our results demonstrate that the LASTS representation can be applied across multiple tasks without any investment needed in designing custom models for each task. Thirdly, LASTS work effectively with multiple LLM adaptation techniques without algorithm specific alternations. Fourthly, we observe that StoP as an adaptation technique outperforms other techniques for most time prediction evaluations, and in all event type prediction evaluations. Finally, we highlight our results on EPIC-KITCHENS dataset, which features very rich textual event descriptions (approximately 20,000). While traditional TPP modeling methods struggle to handle such a large set of classes, our approach effectively models various tasks on this complex dataset.

| Model | Amazon | | Retweet | | Taxi | | Taobao | | StackOverflow | | Breakfast | | MultiTHUMOS | | EPIC-KITCHENS | |
|---|---|---|---|---|---|---|---|---|---|---|---|---|---|---|---|---|
| | M-F1 ↑ | RMSE ↓ | M-F1 ↑ | RMSE ↓ | M-F1 ↑ | RMSE ↓ | M-F1 ↑ | RMSE ↓ | M-F1 ↑ | RMSE ↓ | M-F1 ↑ | RMSE ↓ | M-F1 ↑ | RMSE ↓ | M-F1 ↑ | RMSE ↓ |
| RMTPP | 0.0988 | 0.4780 | 0.3110 | 16.5849 | 0.2969 | 0.3761 | 0.4495 | 0.1338 | 0.0277 | 1.3727 | - | - | - | - | OOM | OOM |
| NHP | 0.1266 | **0.4489** | 0.4128 | **15.6233** | 0.3667 | 0.3995 | 0.4287 | 0.1822 | 0.0559 | 1.3960 | 0.0167 | 116.23 | 0.2861 | 4.8583 | OOM | OOM |
| SAHP | 0.0846 | 0.5491 | 0.2772 | 16.6451 | 0.2780 | **0.3193** | 0.1816 | 0.1347 | 0.0322 | 1.3326 | 0.0023 | 112.85 | 0.0 | 4.5908 | OOM | OOM |
| THP | 0.1414 | 0.4911 | 0.2114 | 16.6440 | 0.3451 | 0.3736 | 0.2734 | 0.1340 | 0.0661 | 1.4054 | - | - | - | - | OOM | OOM |
| AttNHP | 0.1270 | 0.7054 | 0.4210 | 16.8278 | 0.2167 | 0.4072 | 0.1048 | 0.1350 | 0.0475 | 1.3661 | 0.0478 | 108.41 | 0.0809 | 5.2113 | OOM | OOM |
| **LASTS + StoP** | **0.1520** | 0.6000 | **0.4299** | 16.4981 | **0.4174** | 0.3278 | **0.4633** | **0.1321** | **0.0983** | **1.2596** | **0.2633** | 102.02 | **0.3947** | 3.6722 | **0.0797** | 7.3724 |

Table 2: Performance of models on next-event's type and type prediction across five real datasets. The **best result** is shown in bold, and the second best result is underlined. OOM indicates an Out Of Memory error. A missing entry indicates the model diverged. We tried optimizing these baselines for the three textual datasets—MultiTHUMOS (65 classes), Breakfast (177 classes), and EPIC-KITCHENS (∼ 20K classes)—but these models either diverged, performed poorly, or ran out of memory due to the large number of classes.

**Comparison with TPP models.**    Table 2 shows experimental results that compare our model with existing TPP models on standard TPPs datasets. TPP models are designed for forecasting so we only show the results for the forecasting task. We observe that our model is having competitive results w.r.t. TPP models. Our model is outperforming existing TPP models on 13 of the 18 evaluations, and is in the top-2 best models on 17 of the 18 evaluations. Our model has the best performance for all the event type evaluations, which shows that our model is more accurate to predict the next event type. On 3 of the 8 datasets, our model is less accurate than TPP models to predict the time. We think that our model is not performing as well as the TPP models, because our model does not have explicit prior about the time distribution whereas TPP models make strong assumptions about the time distribution (*e.g.* Poisson process or Hawkes process). In the case of the Amazon dataset, the performance gap is more pronounced because this dataset groups a large number of diverse event types into a single event category, making it harder to model inter-arrival times. These results show that our model is able to outperform existing TPP models on most of the datasets without explicit modeling of the time distribution. We think it may be possible to improve the performance of our model by adding a distribution prior in the prompt, and leave it as future work. It also shows that our model is performing well even when only the index of the event type is provided instead of its textual description, making it a more generally applicable method. See Appendix A.5 for further discussion.

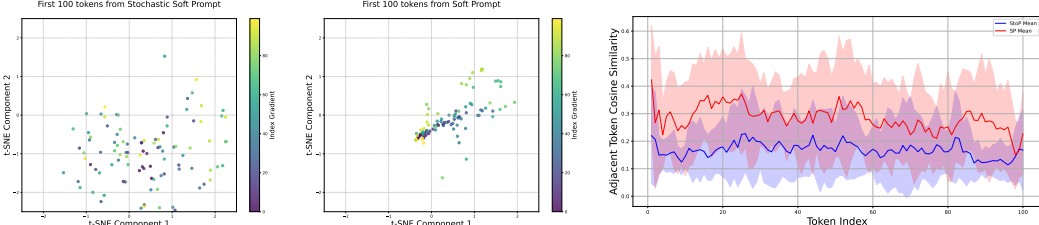

Figure 4: Analysis of learned token representations of Stochastic Soft Prompt (StoP) and Soft Prompt (SP): The first two plots show t-SNE projections of the first 100 tokens from 400-length StoP and SP prompts respectively, trained on the Breakfast dataset for forecasting. StoP tokens are more dispersed, while SP tokens are closely clustered. The third plot shows cosine similarity between adjacent tokens for SP (red) and StoP (blue) across multiple prompts and datasets, with lower similarity for StoP, indicating greater diversity.

## 5.3  MODEL ANALYSIS

**Comparison of SP and StoP learned token representations.**    The tokens learned by Stochastic Soft Prompt (StoP) and Soft Prompt (SP) have distinct characteristics due to differences in their training paradigms. To illustrate this difference, we plot the t-SNE projections of the first 100 tokens from a prompt of length 400 for both StoP and SP in Figure 4. We observe that the tokens learned through StoP training are more spread out, indicating greater diversity, while those learned through SP training tend to cluster more closely. StoP uses a coarse-to-fine approach, where the first embeddings are more diverse to cover a large part of the space than the first embeddings trained with SP. This difference is further highlighted by the cosine similarity between adjacent tokens in the last plot of Figure 4: the adjacent tokens in StoP prompts have lower similarity compared to SP. It allows StoP to work better than SP, even when only the first soft tokens are used (see Figure 5). Using more soft tokens further improves StoP, as it gains access to more fine-grained information.

**All prefixes are valid prompts in StoP**    The training paradigm of StoP forces all prefixes of StoP to act as valid standalone prompts, as they are used as prompts during training for some batches (if trained for long enough). (see Figure 5). This further strengthens our belief that tokens in StoP are arranged from coarse, independent tokens at the beginning to tokens with tokens containing finer information towards the end. See Appendix A.10 for further discussion on StoP structure.

**Disentangling stochasticity and prefix picking in StoP.**    To further emphasize that prefix picking during the training regime of StoP is a key contributing factor to the performance improvement, we compare StoP with an alternative training paradigm where, instead of selecting a prefix, we randomly select $l$ tokens from the prompt during each batch, with $l$ drawn from a uniform distribution. This comparison helps to distinguish the effects of introducing stochasticity alone from the struc-

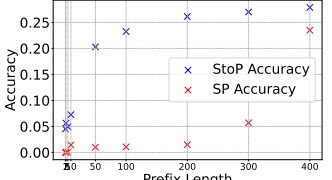
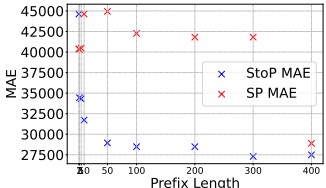

Figure 5: Prefixes of prompts trained using StoP are valid prompts themselves, unlike SP prompts. We take two 400-length prompts—one trained using StoP and the other using SP on the Breakfast dataset for imputation. We evaluate the performance of their prefixes on the test set to assess their feasibility as standalone prompts. The results show that StoP prefixes act as valid prompts, whereas SP prefixes do not.

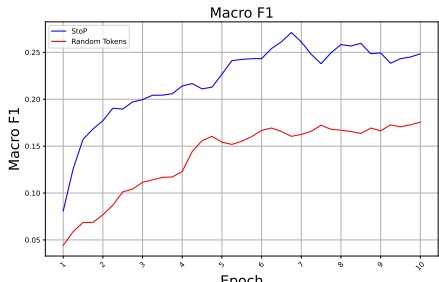
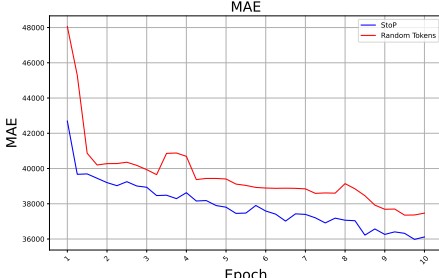

Figure 6: Comparison of Macro-F1 and MAE for StoP vs. random token selection during training, evaluated on validation data after 10 epochs. Results show that random token selection fails to learn effective prompts, while StoP's structured prefix selection achieves significantly better performance.

tured prefix picking employed by StoP. Figure 6 shows a comparison of the macro F1 and MAE metrics on the validation data as both prompts are trained for 10 epochs. These plots show that stochasticity alone is not sufficient for learning good soft prompts, and structured prefix picking is a key component of the StoP training.

**Training speed.** Another dimension to compare SP and StoP is the training speed. Due to differences in training paradigms, StoP trains significantly faster than SP for the same prompt length, as many training batches use only a subset of the full prompt in StoP. In our experiments with 400 soft prompts, we observed that StoP trains approximately 25% faster than SP.

**Understanding StoP prompts through probing** While prior work such as Lester et al. (2021) attempts to interpret learned prompts by mapping them to the closest input embeddings—often yielding incoherent results—we instead explore probing the LLM using the learned prompt. By appending the learned prompt with a simple instruction, such as *"Tell me in as much detail as possible what task you are supposed to do,"* we encourage the LLM to generate an output that reflects its understanding of the task. This approach allows us to gain some insight into what the model has summarized from the tasks and datasets it has been trained on. We present multiple model responses when probed like this in Appendix A.6.

## 6 CONCLUSION AND FUTURE WORK

We explored a novel approach to building an asynchronous time series model using an LLM, offering a new perspective distinct from traditional TPP methods. The method we proposed to encode an asynchronous time series in a prompt suggests that the model can leverage an LLM's world knowledge to perform various downstream tasks such as forecasting, anomaly detection, and imputation.

Furthermore, Stochastic Soft Prompt (StoP), an interpretable adaptation of soft prompt, seems to be efficient in adapting a LLM to asynchronous time series data. We believe this approach could be extended to other data genres like image or natural language texts. We also hope that our findings will open new avenues for research on asynchronous time series models.

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

# A    APPENDIX

## A.1    DATASET PREPARATION

We remove any sequence in the dataset that is very small ($< 4$ elements). We split the dataset in a random $70/10/20$ train, validation and test split. Each sequence is expanded into multiple sequences based on the task:

- **Forecasting**: We convert a sequence into multiple prediction tasks. For each element of the series, the prediction task is to predict the element given the preceding elements. We impose a minimum and maximum length requirements on the number of preceding elements used.

- **Imputation**: For every element in the series, we replace the element by a mask, and the imputation task is to predict the masked element given the remaining sequence.

- **Anomaly Detection**: For every element in the sequence, we replace the action by a random different action. the anomaly detection task is to identify the element of the sequence that has been tampered with.

For the three test based datasets - Breakfast, MultiTHUMOS and EPIC-KITCHENS, the event types are already represented as text. The remaining 5 datasets from the temporal point processes domain lack a textual component, and the event types are represented by integers. For these datasets, we simply treat each integer event type as a string, allowing the LLM to process it similarly to text-based data.

## A.2    LASTS REPRESENTATION OF ASYNCHRONOUS TIME SERIES FOR ZERO SHOT

Here we present the LASTS prompt structure for use with LLMs for various tasks. The structure of the LASTS prompts is shown in Figure 2.

**System Prompt**    The system prompt is very similar across tasks, except for the task specific portions of the prompt. The system prompt used for Forecasting is:

> You are a helpful assistant. Your task is to complete an asynchronous time series. *dataset_description*. Each series is given in the format (inter_arrival_time, action_name). This indicates that the action_name started inter_arrival_times milliseconds after the start of the previous action or the beginning of time if it's the first action. The allowable actions are: *valid_vocab*. Given the first few elements of an asynchronous time series, your task is to provide the next action with its inter arrival time as (inter_arrival_time, action_name). You generate all your response as a single python tuple. Be sure to provide only that one python tuple and nothing else.

The system prompt used for Imputation is:

> You are a helpful assistant. Your task is to find a missing value in an asynchronous time series. *dataset_description*. Each series is given in the format (inter_arrival_time, action_name). This indicates that the action_name started inter_arrival_times milliseconds after the start of the previous action or the beginning of time if it's the first action. The allowable actions are: *valid_vocab*. One of the elements in the series would be missing, marked by the word 'MISSING'. Provide your answer as a single python tuple (inter_arrival_time, action_name) which is your estimate of the missing element of the series. Be sure to give me that one missing python tuple as your response and nothing else.

The system prompt for Anomaly Detection is:

> You are a helpful assistant. Your task is to find an anomolous value in an asynchronous time series. *dataset_description*. Each series is given in the format (inter_arrival_time, action_name). This indicates that the action_name started inter_arrival_times milliseconds after the start of the previous action or the beginning of time if it's the first action. The allowable actions are: *valid_vocab*. One of the elements in the series is an anomaly, and your task is to identify this element which doesn't belong in the series. Provide your answer as a single python tuple (inter_arrival_time, action_name) which is an element from the series you think is an anomaly. Just give me that one anomolous python tuple from the series as your answer and nothing else.

Here, **dataset_description** is a short one line description of the underlying dataset, for example: *"The underlying dataset is derived from tagged human actions while cooking/preparing meals"*.

Also, **valid_vocab** is a comma separated list of allowable action descriptions, if we choose to provide this list and if this list is small.

**User Prompt**  The user prompt in all three tasks is a comma separated string of sequence events, for example

*(0,wait),(139000,carry_bowl),(26000,hold_bowl),*

In case of imputation, there would be a missing element marked by the word MISSING, like so:

*(0,wait),(139000,carry_bowl),MISSING,(41000,reach_eggcarton),*

**Assistant Prompt**  This is empty for zero-shot, as it is filled by the LLM as its prediction for the task on the given sequence.

A.3  EVALUATING LLM INTERACTION WITH LASTS COMPONENTS

We considered various variants of framing the LASTS prompt and present a few interesting ones here, evaluated on Breakfast dataset.

**Testing LLMs use of world knowledge**  We want to test whether LLMs can understand a prompt like LASTS and provide a meaningful response to the task on the sequence using their world knowledge. To this end, we study a variant where each event description is replaced by a uniquely mapped gibberish 4-letter string. This unique mapping ensures that while any semantic meaning in the descriptions is removed, the structure of the time series remains intact. Table 3 shows that all tracked metrics degrade considerably in the scrambled names variant. This confirms that LLMs not only understand LASTS properly but also leverage their world knowledge to perform the specified tasks.

| | Forecast | | | | | |
|---|---|---|---|---|---|---|
| | **M-F1** ↑ | **% Δ** | **Acc** ↑ | **% Δ** | **MAE** ↓ | **% Δ** |
| Zero Shot | 0.0432 | | 0.0866 | | 37.8030 | |
| Scrambled Names | 0.0140 | ↓ -67.63% | 0.0397 | ↓ -54.13% | 38.0742 | ↑ 0.72% |
| | **Imputation** | | | | | |
| Zero Shot | 0.0248 | | 0.0338 | | 33.7669 | |
| Scrambled Names | 0.0100 | ↓ -59.73% | 0.0224 | ↓ -33.73% | 40.4918 | ↑ 19.92% |
| | **Anomaly Detection** | | | | | |
| Zero Shot | 0.0760 | | 0.0650 | | NA | |
| Scrambled Names | 0.0619 | ↓ -18.55% | 0.0469 | ↓ -27.88% | NA | |

Table 3: Comparing LASTS Zero Shot with the Scrambled Names variant across Forecast, Imputation, and Anomaly Detection tasks. Higher values are better for M-F1 and Acc, while lower values are better for MAE. Red indicates negative impact, while green indicates favorable impact.

**Sequence Representation**  We probe about the right representation for the time series events - should they be represented as $(e_i, t_i)$ or $(t_i, e_i)$. Our results in Table 4 show that its better to have time first, followed by the event description. This is what we adopt in LASTS.

| | **Forecast** | | | | | |
|---|---|---|---|---|---|---|
| | **M-F1** ↑ | % Δ | **Acc** ↑ | % Δ | **MAE** ↓ | % Δ |
| Time First $(t_i, e_i)$ | 0.0432 | | 0.0866 | | 37.8030 | |
| Event First $(e_i, t_i)$ | 0.0409 | ↓ 5.38% | 0.0726 | ↓ 16.07% | 37.5344 | ↓ 0.71% |
| | **Imputation** | | | | | |
| Time First $(t_i, e_i)$ | 0.0248 | | 0.0338 | | 33.7669 | |
| Event First $(e_i, t_i)$ | 0.0071 | ↓ -71.30% | 0.0150 | ↓ -55.56% | 31.8194 | ↓ -5.77% |
| | **Anomaly Detection** | | | | | |
| Time First $(t_i, e_i)$ | 0.0760 | | 0.0650 | | NA | |
| Event First $(e_i, t_i)$ | 0.0858 | ↑ 12.94% | 0.0619 | ↓ -4.81% | NA | |

Table 4: Comparison of two ways to express events in an asynchronous time series - event first or time first across Forecast, Imputation, and Anomaly Detection tasks. Higher values are better for M-F1 and Acc, while lower values are better for MAE. Red indicates negative impact, while green indicates favorable impact.

**Time Representation**  We investigate if simplifying the series representation would improve LLM performance. For the Breakfast dataset, we replace inter-arrival times with durations, since we hypothesize that most actions occur contiguously for this dataset. We hypothesize that durations may be easier for the LLM to model rather than inter arrival. From the results in Table 5, we observe that while we have a favourable impact on forecast, both imputation and anomaly detection suffer from this change. This suggests that while durations help with forecasting, more precise inter-arrival times are crucial for more involved tasks like imputation and anomaly detection.

| | **Forecast** | | | | | |
|---|---|---|---|---|---|---|
| | **M-F1** ↑ | % Δ | **Acc** ↑ | % Δ | **MAE** ↓ | % Δ |
| Zero Shot | 0.0432 | | 0.0866 | | 37.8030 | |
| Durations | 0.0600 | ↑ 38.84% | 0.0953 | ↑ 10.12% | 33.781 | ↓ 10.62% |
| | **Imputation** | | | | | |
| Zero Shot | 0.0248 | | 0.0338 | | 33.7669 | |
| Durations | 0.0140 | ↓ -43.56% | 0.0288 | ↓ -14.81% | 29.6881 | ↓ -12.09% |
| | **Anomaly Detection** | | | | | |
| Zero Shot | 0.0760 | | 0.0650 | | NA | |
| Durations | 0.0767 | ↑ 0.96% | 0.0532 | ↓ -18.20% | NA | |

Table 5: Comparison of LASTS Zero Shot with the variant using durations instead of inter-arrival times across Forecast, Imputation, and Anomaly Detection tasks. Higher values are better for M-F1 and Acc, while lower values are better for MAE. Red indicates negative impact, while green indicates favorable impact.

## A.4   LASTS REPRESENTATION USED FOR LLM ADAPTATION

For our experients on LLM adaptation, we keep the LASTS representation very similar to our zero shot experiments:

- **System prompt** in this case is a very concise description of just the task. We skip any dataset description as we expect the model to learn that during the fine tuning process.
- **User prompt** is represented as a comma separated sequence of tuples of event description and inter arrival times.
- **Assistant prompt** contains the expected prediction.

The exact system prompt used for each of the tasks are as follows:

- **Forecasting**: *"Predict the next element of this asynchronous time series where each element is of the form (inter_arrival_time, action_name)."*
- **Imputation**: *"Predict the element marked 'MISSING' in this asynchronous time series where each element is of the form (inter_arrival_time, action_name)."*

- **Anomaly Detection**: *"One of the element in this asynchronous time series is anomalous, find this element. Each element of the series is of the form (inter_arrival_time, action_name)."*

## A.5 BASELINES

**Random Baseline**  To evaluate our methods on the three text-based datasets and the three tasks, we establish a random baseline simulating random guesses. For forecasting and imputation, given an input asynchronous time series, the baseline predicts the inter-arrival time as the average of all inter-arrival times in the sequence and selects a random event type from the valid event descriptions. For anomaly detection, it randomly labels an event from the series as anomalous (see Table 1).

**Foundation Models for Time Series Baseline**  We adapted Chronos (Ansari et al., 2024), a state-of-the-art foundation model designed for zero-shot forecasting on time series data, as a baseline for forecasting and imputation tasks on asynchronous time series datasets. We use the largest model version (*amazon/chronos-t5-large*) available which contains $710M$ model parameters. Since Chronos exclusively handles numerical data, we converted our event descriptions into categorical representations. Each asynchronous time series of length $n$ was transformed into a sequence of $2n$ integers, alternating between inter-arrival times and event categories.

For forecasting, the task was framed as predicting the next two elements in this sequence given the historical context. Adapting Chronos for imputation, however, required additional considerations since it is inherently designed for forecasting. We reformulated the imputation task as a forecasting problem: if the prefix leading up to the missing element is longer than the suffix following it, we treated imputation as forecasting the missing element using the prefix as context. Conversely, if the suffix is longer, we reversed the suffix and used it as context to forecast the missing element. This approach ensures the longest possible context is utilized for predicting the missing value.

It is worth noting that adapting Chronos for anomaly detection is not straightforward, as anomaly detection involves identifying a single anomalous event within the series, which does not align with Chronos' forecasting capabilities. Consequently, Chronos is provided as a baseline exclusively for forecasting and imputation tasks.

**LLMs for Time Series Baselines**  We adapted two LLM-based methods for time series: **LLMTime** (Gruver et al., 2024) and **LLMProcesses** (Requeima et al., 2024), as baselines. Since both methods are designed for numerical time series, we converted textual event descriptions into categorical representations.

LLMTIME  In this method, each data point is represented as a pair: (inter-arrival-time, event-categorical). We modified the default next-token prediction behavior of the model using simple task-specific prompts:

- **Forecasting**: *Predict the next time and event.*

- **Imputation**: *Find the element marked as 'MISSING.'*

- **Anomaly Detection**: *Find the anomalous time and event.*

LLMPROCESSES  This method uses in-context learning with $(x, y)$ examples derived from a sequence, treating the sequence as a real-valued function on a 2D space as domain. In this setup, $x$ represents a point in 2D space $(x_1, x_2)$, where $x_1$ denotes the sequence position, and $x_2$ indicates the output type: 0 for inter-arrival time and 1 for event categorical. For a given sequence, we crafted two distinct prompts: one for predicting the event categorical and another for predicting the inter-arrival time, based on the corresponding value of $x$. We followed the recommended settings from the original paper for prompt construction.

However, anomaly detection does not align with this framework, as it involves identifying a single anomalous time point where the function output is 0 everywhere except at the anomaly. This makes it unsuitable for predicting function values at unseen points based on prior observations. Consequently, we adapted this approach exclusively for forecasting and imputation tasks.

**TPP Models as Baselines** We compare our best fine-tuned model configuration, $LASTS + StoP$, against current state-of-the-art methods for forecasting on asynchronous time series. These methods are adapted from the benchmark study in (Xue et al., 2024). The evaluation spans eight datasets, five of which—Amazon, Retweet, Taxi, Taobao, and StackOverflow contain event categoricals without textual descriptions and are regarded as standard benchmarks for asynchronous time series analysis.

We benchmark the TPP models covered in the EasyTPP benchmark (Xue et al., 2024) on the three textual datasets considered in our work: Breakfast, MultiTHUMOS, and EPIC KITCHEN. Since these datasets represent events as text and TPP models are not equipped to handle text directly, we converted the event names into event categoricals to make them compatible with these models.

**Observations** We summarize our comparison of various baselines with LASTS Zero Shot in Figure 7. We observe that Chronos performs the weakest among the baselines, yet it remains competitive. This is expected as Chronos, while being a much smaller model compared to LLMs, is highly specialized for time series forecasting, which enables it to achieve decent performance. LLMTime and LLMProcesses also perform competitively, especially on the MultiTHUMOS dataset. We attribute this to the noisy nature of the MultiTHUMOS dataset, which includes non-standard event names (e.g., "OneHandedCatch," "TalkToCamera", etc) and repetitive, less meaningful patterns (e.g., "GolfSwing, Wait, GolfSwing, Wait..."). These characteristics may help event-categorical-based models like LLMTime and LLMProcesses. However, on the other two datasets—Breakfast and EPIC_KITCHEN—the textual descriptions of events provide a significant advantage, as evident from the comfortable margin by which LASTS Zero Shot outperforms LLMTime and LLMProcesses across all tasks.

Furthermore, we observed that existing TPP-based models struggled with datasets containing a large number of unique event types, often performing poorly, failing to converge, or encountering out-of-memory errors. This highlights the challenges these models face in handling the diversity and complexity of such datasets.

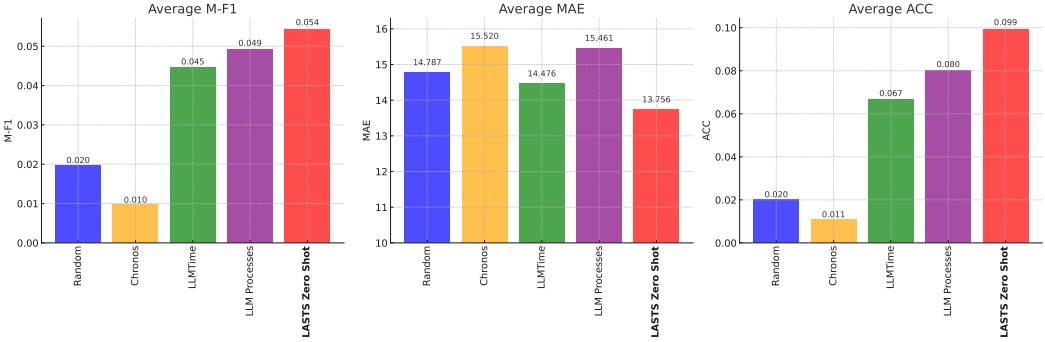

Figure 7: Comparison of performance metrics: Macro-F1 (M-F1), Mean Absolute Error (MAE), and Accuracy (ACC), averaged across all datasets for Forecast and Imputation tasks. Higher values for M-F1 and ACC indicate better performance, while a lower value of MAE is preferred. It is evident that LASTS Zero Shot (our method) achieves the highest average M-F1 and average ACC among all the baselines and also produces the lowest MAE.

A.6 STOP PROMPT INTERPRETATIONS THROUGH MODEL PROBING

Prior work such as Lester et al. (2021) attempts to interpret learned prompts by mapping them to the closest input embeddings. However, this approach often produces sequences that are difficult to understand or lack meaningful content. For example, Xu et al. observed that the closest words to the learned embeddings are mostly meaningless, several tokens are mapped to the same word, and the cosine similarity between the tokens and their closest word embeddings almost always falls below 0.16. Instead, we explore probing the LLM using the learned prompt. By appending the learned prompt with a simple instruction, such as *"Tell me in as much detail as possible what task you are supposed to do,"* we encourage the LLM to generate an output that reflects its understanding of the task. This approach allows us to gain meaningful insights into what the model has summarized from

the tasks and datasets it has been trained on. Here we present a few interesting interpretations of StoP prompts obtained by model probing.

**StoP of Length** 400**, Breakfast Dataset, Forecast**   We obtain a very nice summarization of the breakfast dataset:

> The task appears to be related to preparing a snack or meal by using the facilities provided in the kitchen. Specifically, I was asked to perform a series of actions to accomplish this goal. Here's what I've observed:1. **Opening the cabinet**: I need to open the cabinet to access the necessary utensils, cookware, or ingredients.2. **Grabbing the ingredients**: I will need to pick up the specified ingredients, such as a bowl, a cutting board, or a can opener, to prepare the desired dish.3. **Slicing the ingredients**: I will cut the ingredients into the required proportions, shapes, or sizes depending on the recipe being followed.4. **Preheating the cooking surface**: If needed, I will turn on the stove, oven, or other cooking device to the desired temperature to start cooking the ingredients.5. **Cooking the ingredients**: I will proceed to cook the ingredients according to the required cooking method, such as boiling, saut´eing, frying, or baking.6. **Adding condiments or seasonings**: If necessary, I will add the required condiments, spices, or seasonings to the cooked ingredients to enhance their flavor.7. **Combining the ingredients**: Finally, I will combine the cooked ingredients in the desired manner, whether it's mixing, blending, or assembling the final dish.Please let me know if I'm correct in my understanding or if there's anything else I can do to help!

**StoP of Length** 5**, Breakfast Dataset, Forecast**   We obtain very cooking adjacent prompt interpretations from even a small prompt of length 5 trained for forecasting on Breakfast. For example:

> Here is my main task: "Reach for a tablespoon and move in the direction of a plate where I already placed some eggs and a knife. Slowly pour one egg into the plate, take the knife, and carefully cut the egg in halves. Transfer both halves away from the plate, and then carry the remaining plate towards a sink. Turn the tap on, wash the plate, and then move away from the sink, holding the plate still under running water. Remove the plate from the sink, carry it to a countertop, and put it down. Finally, reach for a knife and move towards a bowl containing a knife, carry the knife, and move towards a counter where I already placed a cutting board, where there is a set of utensils lying on it. Take the knife, push towards the board, and reach for a knife. Move away from the cutting board.

**StoP of length** 400**, MultiTHUMOS Dataset, Imputation**   The interpretation seems to have mapped the idea of an asynchronous time series to another domain all together, while still keeping elements like walk, run, stand, etc from the original dataset:

> I'm an AI designed to generate text based on the sequence of events occurring in the events diagram. Specifically, I´m trained to analyze the sequence of events and identify the corresponding actions or actions sequences in the diagram, and then use that information to generate a textual description of what is happening in the diagram.The main tasks I was designed to perform are:1. **Identify sequence of events**: I analyze the sequence of events in the diagram, identifying the time-stamped data points and the transitions between them.2. **Match events with actions**: I match the identified sequence of events with the corresponding actions or actions sequences. For example, if I see a sequence of time-stamped data points indicating a person sitting, standing, and then walking, I would match those events to the action of "walking".3. **Generate textual description**: Using the matched events and actions, I then generate a textual description of what is happening in the diagram. For example, if I match the sequence "sitting, standing, walking" to the action "walking", I would generate a text describing that sequence as "walking". Additionally, I also make use of some contextual information, such as the diagram´s caption, to disambiguate and refine my generated textual descriptions. The final textual output is a natural language description of the content of the diagram.

**StoP of length** 400, **MultiTHUMOS Dataset, Anomaly Detection** The interpretations center around games, which is expected as MultiTHUMOS is a gaming dataset. Here are two sampled interpretations:

> The task is to simulate a volleyball game. The events include:1. Slices or spikes the ball with the palm of the hand (VolleyballSpiking). 2. Blocks the ball with the palm of the hand (VolleyballBlock).3. Sets the ball with the palm of the hand (VolleyballSet). 4. Pounds or slams the ball with the fist (VolleyballSpiking). 5. Jumps or reaches for the ball (VolleyballJumping). 6. Moves sideways or slides on the ground (VolleyballSpikeRun).It's important to remember that volleyball spiking, blocking, setting, and jumping involves movement and body language, which can't be reflected directly in a sequential diagram

> The task you are supposed to do is sports broadcasting, and specifically, you are a golf broadcaster. As a golfer, you are supposed to perform a live broadcast of the golf match, providing commentary and analysis as the golfers participate in the match. The situation is simulated, as there are no external stimuli or real golfers, you have to act as both commentator and golfer, making mock commentary and mimicking golf swings.The task requires you to stay in a fixed position, usually behind a microphone, and maintain a semi-serious, professional tone. Your task is to articulate the events on the course, such as drives, putts, and other golf shots, as if you are actually seeing them happen. You should also describe the terrain, distance, and obstacles faced by the golfer, and offer your expert analysis and opinions on their technique, strategy, and chances of scoring.Your commentary should be smooth, clear, and engaging, drawing the audience into the game and painting a vivid picture of the action on the course. Your performance should exude confidence and authority, as if you are an experienced commentator who has watched many matches before. The audience should feel like they are right there on the course with you, experiencing the excitement and suspense of the tournament.

**StoP of Length** 400, **Taobao Dataset, Forecast** Note that this dataset is not textual, so all the soft prompt has seen is the task description and a sequence of pairs of numbers.

*Prompt:* Describe your task.
*Model Response*:

> I'm an AI assistant, my task is to generate a sequence based on a rule that I've learned from given inputs (or "train" sequences). The rule is inferred by analyzing the pattern of increasing or decreasing values of the input sequence. Each sequence is either called ("silenced") or activated, and if there's no change in the sequence, the "silence" is used, unless the input sequence calls the "silence" by itself. The task involves predicting the next output in a sequence, given some input sequences that establish a pattern.

A.7 COMPARIOSN OF LASTS + STOP WITH OTHER PEFT TECHNIQUES

In this section, we compare the performance of LASTS + StoP with other PEFT techniques listed in Table 1. Table 6 highlights the percentage improvements observed in various metrics when using Stochastic Soft Prompting compared to standard Soft Prompting. We observe a significant advantage of Stochastic Soft Prompting across all datasets and tasks, with an overall average increase of 12.69% in Macro-F1 across all tasks and datasets. Similarly, Table 7 demonstrates an average increase of 13.55% in Macro-F1 when using Stochastic Soft Prompting instead of finetuning techniques like QLORA.

| Task | Breakfast | | | MultiTHUMOS | | | EPIC_KITCHEN | | |
|---|---|---|---|---|---|---|---|---|---|
| | M-F1 | MAE | ACC | M-F1 | MAE | ACC | M-F1 | MAE | ACC |
| Forecast | 11.09% | 0.91% | 4.87% | 6.08% | 0.77% | 0.04% | 2.13% | -4.91% | 3.52% |
| Imputation | 23.40% | 2.22% | 17.37% | 7.64% | 2.76% | 10.95% | 30.66% | 1.09% | 10.81% |
| Anomaly Detection | 10.40% | — | 12.82% | 15.56% | — | 10.97% | 7.21% | — | 6.06% |
| **Avg Gain (Per Task)** | 14.96% | 1.56% | 11.69% | 9.76% | 1.76% | 7.32% | 13.33% | -1.91% | 6.80% |
| **Avg Gain (All Tasks, All Datasets)** | **M-F1**: 12.69% | | | **MAE**: 0.47% | | | **ACC**: 8.60% | | |

Table 6: Comparison of LASTS+StoP with LASTS+SP. The table shows the percentage improvement in each metric achieved by using Stochastic Soft Prompting compared to standard Soft Prompting. Significant gains are observed across all datasets and tasks with Stochastic Soft Prompts. On average, across all datasets and tasks, Macro F1 increases by 12.69%.

| Task | Breakfast | | | MultiTHUMOS | | | EPIC_KITCHEN | | |
|---|---|---|---|---|---|---|---|---|---|
| | M-F1 | MAE | ACC | M-F1 | MAE | ACC | M-F1 | MAE | ACC |
| Forecast | 2.93% | 4.39% | 3.11% | 22.65% | 4.71% | 10.31% | 4.32% | -4.47% | 6.39% |
| Imputation | 22.27% | 1.20% | 9.60% | 3.80% | -5.40% | 3.46% | 61.38% | 0.25% | 25.24% |
| Anomaly Detection | 2.67% | — | 3.40% | 0.70% | — | 1.65% | 1.27% | — | 0.70% |
| **Avg Gain (Per Task)** | 9.29% | 2.79% | 5.37% | 9.05% | -0.34% | 5.14% | 22.32% | -2.11% | 10.78% |
| **Avg Gain (All Tasks, All Datasets)** | **M-F1**: 13.55% | | | **MAE**: 0.11% | | | **ACC**: 7.10% | | |

Table 7: Comparison of LASTS+StoP with LASTS+QLORA. The table shows the percentage improvement in each metric achieved by using Stochastic Soft Prompting compared to finetuning via QLORA. Significant gains are observed across all datasets and tasks with Stochastic Soft Prompts. On average, across all datasets and tasks, Macro-F1 increases by 13.55%.

| | # Params | Breakfast | | MultiThumos | | EPIC KITCHEN | |
|---|---|---|---|---|---|---|---|
| | | Macro F1 ↑ | MAE ↓ | Macro F1 ↑ | MAE ↓ | Macro F1 ↑ | MAE ↓ |
| Forecast | 1B | 0.2292 | 33.9309 | 0.3210 | 1.8013 | 0.0574 | 3.0859 |
| | 3B | 0.2526 | 33.2541 | 0.3694 | 1.7259 | 0.0708 | 3.0169 |
| | 8B | 0.2633 | 32.5464 | 0.3947 | 1.6503 | 0.0797 | 3.0318 |
| Imputation | 1B | 0.0256 | 31.1075 | 0.0907 | 2.4256 | 0.0102 | 3.2571 |
| | 3B | 0.0966 | 31.1597 | 0.1329 | 2.3963 | 0.0280 | 3.1445 |
| | 8B | 0.2064 | 28.2251 | 0.2213 | 2.3445 | 0.0610 | 3.1116 |
| Anomaly Detection | 1B | 0.0688 | — | 0.0954 | — | 0.0318 | — |
| | 3B | 0.5726 | — | 0.4777 | — | 0.5793 | — |
| | 8B | 0.7198 | — | 0.6045 | — | 0.6603 | — |

Table 8: Comparison of Macro-F1 and MAE across the Breakfast, MultiThumos, and EPIC_KITCHENS datasets for forecasting, imputation, and anomaly detection as the number of model parameters varies. The results show that Macro-F1 consistently improves with increasing model size across all datasets and tasks. In most cases, MAE decreases as model size increases, confirming that larger models generally lead to better performance.

## A.8    SCALING TO DIFFERENT LLM BACKBONE SIZES

We trained Stochastic Soft Prompts (StoP) across different backbone sizes of large language models and observed consistent improvements in performance as the model size increased. Specifically, we conducted experiments using LLama3.2 models with 1B and 3B parameters, as well as the LLama3-8B Instruct model. These improvements were clear across the Breakfast, MultiThumos, and EPIC_KITCHENS datasets and applied to all tasks - forecasting, imputation, and anomaly detection.

Notably, Table 8 and Figure 8 show that macro-F1 scores consistently improve with larger model sizes across all datasets and tasks. Additionally, Mean Absolute Error (MAE) decreased in most cases as the model size increased, further confirming that larger models help Stochastic Soft Prompts perform better by utilizing their enhanced representational power. The performance difference be-

tween model sizes is smaller for forecasting tasks since these align with the next-token prediction that LLMs are trained on. However, for harder tasks like imputation and anomaly detection, the improvements are much larger as model size increases.

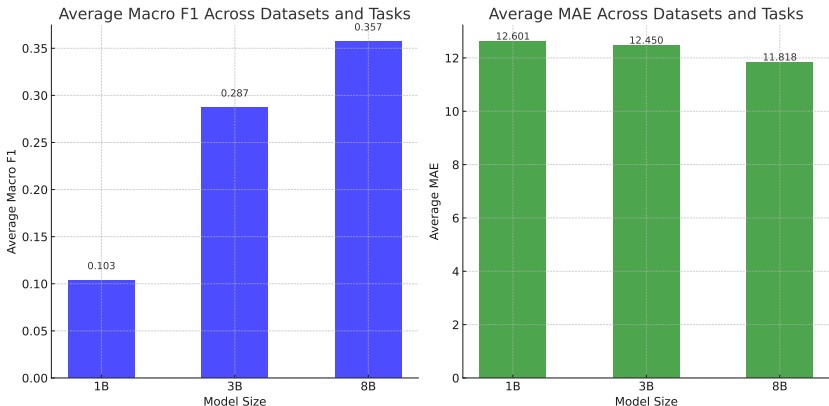

Figure 8: Comparison of average Macro F1 and MAE across all datasets and tasks for different model sizes. The left histogram shows the average Macro F1 scores, while the right histogram depicts the average MAE values. We see a clear trend of improvement in both metrics as model sizes increase.

| Few-Shot (k) | Breakfast | | | MultiTHUMOS | | | EPIC-KITCHENS | | |
|---|---|---|---|---|---|---|---|---|---|
| | M-F1 ↑ | MAE ↓ | ACC ↑ | M-F1 ↑ | MAE ↓ | ACC ↑ | M-F1 ↑ | MAE ↓ | ACC ↑ |
| | | | | | Forecast | | | | |
| $k = 0$ | 0.0604 | 38.1630 | 0.0969 | 0.1361 | 1.8868 | 0.1826 | 0.0105 | 3.1566 | 0.0920 |
| $k = 1$ | 0.1312 | 37.6239 | 0.1808 | 0.1393 | 1.7913 | 0.2381 | 0.0144 | 3.2606 | 0.1123 |
| $k = 2$ | 0.1257 | 36.4688 | 0.1870 | 0.1622 | 1.7960 | 0.2505 | **0.0151** | 3.2266 | **0.1180** |
| $k = 5$ | 0.1518 | **35.5605** | 0.2133 | 0.1676 | 1.8114 | 0.2581 | 0.0149 | 3.3092 | 0.1150 |
| $k = 7$ | 0.1491 | 35.6785 | 0.2107 | **0.1991** | **1.7810** | **0.2828** | 0.0138 | 3.2177 | 0.1002 |
| $k = 10$ | **0.1667** | 37.6084 | **0.2442** | 0.1807 | 1.7820 | 0.2397 | 0.0124 | **3.0904** | 0.0901 |
| | | | | | Imputation | | | | |
| $k = 0$ | 0.0263 | 33.0097 | 0.0594 | 0.0915 | 2.6696 | 0.1210 | 0.0015 | 3.6527 | 0.0446 |
| $k = 1$ | 0.0419 | 33.1403 | 0.0738 | 0.1165 | 2.5106 | 0.1490 | 0.0018 | 3.6402 | **0.0569** |
| $k = 2$ | **0.0527** | **31.1138** | 0.0826 | 0.1102 | **2.3576** | 0.1486 | 0.0022 | 3.5375 | 0.0527 |
| $k = 5$ | 0.0520 | 33.3440 | 0.1001 | 0.1013 | 2.3982 | **0.1569** | **0.0023** | **3.2528** | 0.0547 |
| $k = 7$ | 0.0509 | 34.0198 | 0.0994 | 0.1001 | 2.4228 | 0.1462 | 0.0019 | 3.3447 | 0.0475 |
| $k = 10$ | 0.0474 | 31.2001 | **0.1069** | **0.1219** | 2.3771 | 0.1546 | 0.0015 | 3.2552 | 0.0406 |
| | | | | | Anomaly Detection | | | | |
| $k = 0$ | 0.0923 | — | **0.0763** | 0.2755 | — | 0.1949 | 0.0159 | — | 0.0777 |
| $k = 1$ | 0.1002 | — | 0.0681 | 0.2809 | — | 0.1961 | 0.0172 | — | 0.0854 |
| $k = 2$ | 0.0739 | — | 0.0569 | 0.3361 | — | **0.2891** | 0.0213 | — | 0.1062 |
| $k = 5$ | 0.0837 | — | 0.0563 | **0.3535** | — | 0.2720 | **0.0337** | — | **0.1637** |
| $k = 7$ | 0.0705 | — | 0.0469 | 0.3436 | — | 0.2516 | 0.0278 | — | 0.1369 |
| $k = 10$ | **0.1026** | — | 0.0700 | 0.2340 | — | 0.1629 | 0.0222 | — | 0.1097 |

Table 9: Comparison of performance metrics (M-F1, MAE, and ACC) across Breakfast, Multi-THUMOS and EPIC_KITCHEN datasets over forecast, imputation and anomaly detection tasks for different few-shot values $k$ given as in context examples. $k = 0$ indicates Zero Shot. Higher M-F1 and ACC values indicate better performance, while lower MAE values are better. MAE computation is not applicable for anomaly detection. Best metric values are indicated in **bold**.

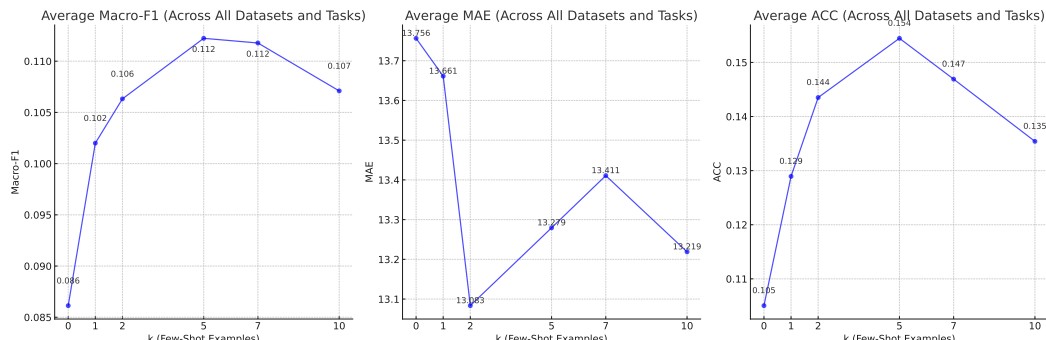

Figure 9: Average values of Macro-F1, MAE, and ACC across all datasets and tasks for different values of $k$ (number of few-shot examples). Higher values indicate better performance for Macro-F1 and ACC, while lower values indicate better performance for MAE. The results indicate that on an average, $k = 5$ works best.

### A.9 LASTS FEW SHOT

We study the impact of varying the number of examples ($k$) in the few-shot setting to determine the optimal value of $k$ for our method. Specifically, we evaluate the performance of LASTS Few Shot on all datasets and tasks using different $k$ values, ranging from $k = 0$ (Zero Shot) to $k = 10$. As shown in Figure 9 and detailed in Table 9, the performance metrics—Macro-F1, MAE, and ACC—improve significantly as $k$ increases from 0 to 5. However, further increases in $k$ beyond 5 do not consistently yield improvements and, in some cases, result in marginal performance degradation.

On average, $k = 5$ achieves the best balance across all metrics and datasets. Therefore, we adopt $k = 5$ as the default value for LASTS Few Shot and include it as the entry for "LASTS Few Shot" in Table 1.

### A.10 FURTHER ANALYSIS ON STOCHASTIC SOFT PROMPTS (STOP)

In this section, we comment on the structure learned by StoP prompts and discuss the practical benefits of Stochastic Soft Prompts.

**Evidence for Coarse-to-Fine Structure** The prompts learned through Stochastic Soft Prompts (StoP) suggest the presence of a structured coarse-to-fine hierarchy. In this structure, the first few tokens appear to encode broader task-level information, while later tokens may refine predictions by adding more detailed nuances. Below, we provide observations that support this behavior:

1. **t-SNE Projections:** Visualizations of t-SNE projections (see Figure 10) suggest that the first few tokens in StoP prompts may encode more diverse or independent representations, as indicated by their wider spread in the projection space. In contrast, the later tokens tend to cluster more closely together, potentially reflecting the refinement of previously encoded information.

2. **Cosine Similarity:** Adjacent tokens at the beginning of the StoP prompt tend to exhibit lower cosine similarity compared to tokens later in the prompt (see Figure 10). This pattern suggests more diverse information being captured at the beginning of the prompt. Standard soft prompts, however, show uniformly high cosine similarities across all tokens, lacking this structure.

3. **Prefix Validity:** Figure 5 indicate that any prefix of a StoP prompt serves as a valid standalone prompt, with additional tokens refining the predictions. This behavior suggests that early tokens convey broad task-level information, while later tokens refine and add finergrained details.

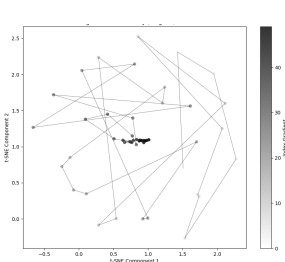 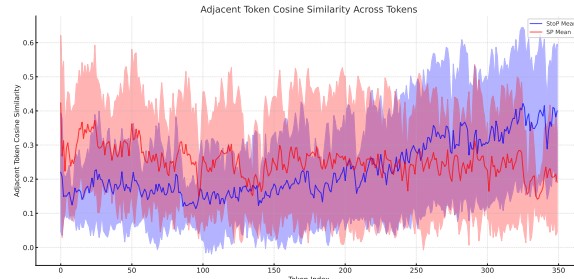

Figure 10: *Left*: t-SNE projections of Stochastic Soft Prompt (StoP) tokens with a prompt length of 50 on the Breakfast dataset for the forecasting task. Adjacent tokens are connected by a line, and the color darkens as the token index increases. The presence of lighter tokens on the periphery and darker tokens in the center indicates that the initial tokens learn very diverse information, while this diversity diminishes as the token index increases. *Right*: Pairwise cosine similarity of the first 350 tokens of a stochastic soft prompt and a soft prompt learned on the Breakfast dataset for forecasting. We observe that in StoP, the initial cosine similarities are smaller and increase as the token index increases, while no such variation by token index is present in a normal soft prompt.

**Practical Benefits of StoP**   We observe that StoP offers many benefits over standard soft prompting:

1. **Improved Generalization:** StoP prompts achieve better generalization compared to standard soft prompts, with an average improvement of **12.69%** in Macro-F1 across all datasets (Breakfast, MultiTHUMOS, and EPIC_KITCHENS) and tasks (Forecast, Imputation, Anomaly Detection) (see Table 6)

2. **Faster Training:** The stochastic nature of StoP reduces training time by approximately **25%**, making it more efficient than standard soft prompting.

3. **Resource Efficiency:** StoP enables flexible deployment in resource-constrained environments. Longer trained StoP prompts can be truncated to prefixes as needed, allowing for adaptable inference without compromising performance.

