# OpenReview forum: "Modeling Asynchronous Time Series with Large Language Models"
_ICLR.cc/2025/Conference — Submitted to ICLR 2025_

### Official Review · Reviewer_aAMq · 2024-10-30

**Soundness:** 3
**Presentation:** 3
**Contribution:** 3
**Rating:** 5
**Confidence:** 3

**Summary:**

This paper proposes a new approach to model asynchronous time series with LLMs which solves three different tasks: forecasting, imputation, and anomaly detection. First, they explored the representations of the asynchronous time series as inputs to LLMs. Second, they studied different parameter-efficient techniques to adapt an LLM for modeling asynchronous time series. The proposed framework achieves competitive performance across different temporal event benchmarks.

**Strengths:**

1. This paper is the first to propose using LLMs for asynchronous time series data. This task has promising research prospects and the experimental results are also encouraging.
2. This paper designs a text-based representation of asynchronous time series for LLMs and explores mainstream parameter-efficient fine-tuning methods on this basis. The results on different datasets demonstrate the effectiveness of the proposed framework.

**Weaknesses:**

1. Although the model in this paper has shown a significant performance improvement in the selected dataset, there is a concern that the performance of the model on all tasks seems to be low (Not sure whether it exceeds or approaches the human level, and it is relatively easy for people to predict daily events, event imputation, or detect event anomalies). This makes me worry that the current task definition is not perfect enough, whether it is too difficult for the model, or usable enough.
2. One of the main experiments in the paper (Table 1) lacks more credible baselines. The author mainly compares with the random guess, which is often not a particularly credible or competitive comparison target. Finding more baselines will help reflect the performance of the method.
3. Scaling laws have been widely demonstrated in LLMs, and I noticed that this paper uses Llama-3-8B-Instruct as a base model, so I was curious whether this approach would generalize to larger models.

**Questions:**

Please see the weaknesses.

**Details Of Ethics Concerns:**

There is no ethics concern needed.

---

> ### Author Response · Authors · 2024-11-21
> **We sincerely thank the reviewer for the detailed comments. We carefully address each of the reviewer’s concerns below and hope that our response resolves your concerns. Any follow-up questions are welcome.**
>
> **W:Although the model in this paper has shown a significant performance improvement in the selected dataset, there is a concern that the performance of the model on all tasks seems to be low … the current task definition is not perfect enough, whether it is too difficult for the model, or usable enough.**
>
> Thank you for your feedback regarding the performance levels of our model. Asynchronous time series (AST) are inherently challenging due to irregular timing and sparsity of events. Most existing studies focus on datasets with a small number of event types, each presenting different levels of difficulty. In our Table 2, we reference datasets like Amazon, Retweet, Taxi, Taobao, and StackOverflow, which have been extensively studied in AST forecasting and are included in benchmarks such as EasyTPP [3]. These datasets are known for their complexity; for instance, prior to our work, the state-of-art Macro-F1 score for the StackOverflow dataset was as low as *0.0661*, underscoring the difficulty of achieving high performance on this task
>
> Studying these challenging datasets is important as they represent real-world scenarios where event occurrences are irregular and sparse. Our contributions aim to advance the field by:
>
> - **Enhancing Traditional Datasets**: We have expanded upon traditional AST datasets by introducing datasets with larger numbers of event types, increasing task complexity and relevance.
> - **Establishing Baselines for Future Research**: By providing baseline results on these more complex datasets, we enable future work to build upon our findings.
>
> We hope this addresses your concern.  We would be happy to provide further clarifications.
>
> **W: One of the main experiments in the paper lacks more credible baselines… Finding more baselines will help reflect the performance of the method.**
>
> Thank you for your feedback on the need for more credible baselines. We have addressed this in Global Question 1 with the addition of more LLM-TS baselines. Please let us know if we can provide further clarification.
>
> **Q: Scaling laws have been widely demonstrated in LLMs, …  whether this approach would generalize to larger models**
>
> While this was listed as a weakness, it sounded to us more like a question. If that is the case, we appreciate it if you can clarify. Our method is not specific to small models; we used Llama-3-8B-Instruct primarily due to computational resource constraints. We fully expect that our approach would generalize well to larger models if more computational resources were available. In fact, *Reviewer 8vh2* also noted that *"this technique will continue to improve as LLMs improve, and as in-context learning techniques improve."* To explore this further, we conducted additional experiments with smaller models, specifically Llama3.2-1B and Llama3.2-3B. We observed a consistent trend of increasing performance as the model size increases, supporting the idea that our approach scales positively with model capacity. We have included these new results and analyses in Appendix A.8 of our manuscript. These findings suggest that our method would likely benefit from even larger models, reinforcing its generalizability and potential for improved performance with access to more powerful language models.

---

> > ### Comment · Reviewer_aAMq · 2024-11-26
> >
> > I appreciate the authors' response. I am pleased to see that most of my concerns have been addressed. I would be inclined to raise my score if the authors could incorporate additional competitive baseline experiments and provide corresponding analyses before the rebuttal concludes.

---

> > > ### Author Response · Authors · 2024-11-26
> > >
> > > Thank you for your feedback. We have incorporated **three additional baselines** into our work. The details of these added baselines are covered in the global response above and further elaborated in Section 5.2 and Appendix A.5 of our paper.

---

### Official Review · Reviewer_gifc · 2024-11-03

**Soundness:** 3
**Presentation:** 2
**Contribution:** 2
**Rating:** 3
**Confidence:** 4

**Summary:**

This paper proposes a novel prompt learning framework to leverage the LLM's world knowledge and capability to model async time series.  A stochastic soft prompting is designed to achieve this. Overall, this paper proposes a method on LLM for time series, an interesting topic. But it lacks enough quality from many aspects to be accepted.

**Strengths:**

1. This is a good topic. LLM for time series is still a very promising direction. There are still many research questions to be answered on how to effectively leverage LLM for time series data. This paper designs a prompting method to use LLM for various time series tasks, including forecasting, anomaly detection, and imputation.

**Weaknesses:**

1. "Related work lacks LLM for TS". In the related work section, there is a lack of discussion of LLM for TS. There is already some related work. For example, https://arxiv.org/abs/2402.01801 this survey contains a lot of them.
2. "Performance is not competitive." From Table 1, the StoP prompt learning method is not significantly better than QLORA.
2. "Lacks comparison with related work." There lacks a comparison with other LLM for TS methods. Currently, this paper only compares their backbone with different prompting strategies.
2. "Figure 1 lacks many details." Figure 1 should present more details of the framework.
3. "Written is weak." The author should improve the writing quality. There are many places hard to understand. For example, the abstract is hard to understand.

**Questions:**

See details in weakness.

---

> ### Author Response · Authors · 2024-11-21
> **We sincerely thank the reviewer for the detailed comments. We carefully address each of the reviewer’s concerns below and hope that our response resolves your concerns. Any follow-up questions are welcome.**
>
> **W: Related work lacks LLM for TS**
>
> Thank you for highlighting the gap in discussion on Large Language Models (LLMs) for time series in our related work section. We have revised the subsection titled "LLMs for Time Series," in the related work section to include a more comprehensive exploration of LLMs applied to time series analysis.
>
> **W: Performance is not competitive.**
>
> Thank you for your feedback regarding the performance comparison between StoP prompt learning and QLoRA. However, we respectfully disagree - as detailed in our results, StoP leads to substantial improvements across various tasks and datasets:
>
> - StoP outperforms QLoRA in **Forecasting** by **+9.97% Macro-F1 (MF1),** in **Imputation** by **+29.15% MF1**, and in **Anomaly Detection** by **+1.55% MF1,** when averaged over our three text based datasets - Breakfast, MultiThumos, Epic Kitchen.
> - Similarly, when averaged over all datasets and all tasks, StoP outperforms QLoRA by  **+13.55% MF1**.
>
> We will include a detailed breakdown of these results in Appendix A.7 before the end of rebuttal period. These results prove StoP's competitive performance across diverse settings. We hope this addresses your concern and provides clarity on the advantages of StoP over QLoRA.
>
> **W: Lacks comparison with related work.**
>
> Thank you for your feedback on the need for more credible baselines. We have addressed this in Global Question 1 with the addition of more LLM-TS baselines. Please let us know if we can provide further clarification.
>
> **W: Figure 1 lacks many details.**
>
> Thank you for your feedback regarding Figure 1. We have updated the figure to clarify that it focuses on the tasks explored in our paper, while details about the framework are presented in Figure 2. We are happy to make additional adjustments based on your suggestions.
>
> **W: Written is weak.**
>
> Thank you for your feedback regarding the writing quality. We were surprised to read this after reviewer 8vh2 praised the paper in this regard *“At a meta level, this paper’s strongest feature is how well it was written. I wish that more AI papers were written with this much clarity and intention.”* Based on your comment, we have revised the abstract to make it clearer and have clarified the language in several portions of the paper. We would be happy to make further improvements if you could point out specific areas that remain unclear. Thank you for your feedback.

---

### Official Review · Reviewer_8vh2 · 2024-11-04

**Soundness:** 4
**Presentation:** 4
**Contribution:** 3
**Rating:** 8
**Confidence:** 4

**Summary:**

This paper considers the problem of asynchronous time series modeling (specifically--the three tasks of forecasting, anomaly detection, and imputation). They take an in-context-learning approach to solving this task. Their main contributions:
1. They propose "LASTS" (Large Language models with Asynchronous Time Series data), a prompt-engineering based method which allows LLMs to solve the asynchronous time series modeling problems in a zero-shot manner.
2. They propose "StoP" (STOchastic soft Prompting), an interpretable adaptation of soft prompting, as part of their prompt engineering strategy. This method involves randomly truncating the soft prompts, which lets the model learn more diverse representations.

**Strengths:**

1. At a meta level, this paper's strongest feature is how well it was written. I wish that more AI papers were written with this much clarity and intention. Overall, I would say the paper is structured in such a clear way that it is easy to evaluate the quality of the underlying research, because as a reader I didn't need to get bogged down in trying to understand what was written.

    * Example 1: , the related works section was truly a delight to read, and I felt like it was very thorough (modulo one missing class of works, see Weakness #2). I especially liked all the different hierarchically-organized categories.
    * Example 2: the background section does a very clear job explaining, in precise mathematical terminology, what the problems being solved are, with minimal notations being introduced (and I know this isn't always easy).
    * Example 3: section 4.2, which explains the background on low-rank adaptation and how it's used in the paper, then soft prompting, then how the paper uses Stochastic soft prompting, is a master class in clearly explaining the background methodology, and how it's used in the present work. This makes this paper very self-contained and clear.

2. Previous works on asynchronous time series typically modeled events as categories, but this paper models them using natural language descriptions (c.f. Section 4.1). This is a clear improvement in flexibility, especially given that the downstream performance improves as well. As a result, it is clear that the authors have proposed a superior framework. Furthermore, the use of ICL with LLMs means that this technique will continue to improve as LLMs improve, and as ICL techniques improve.

3. The results in Table 1 provide a very clear ablation, showing that the proposed ICL-based strategy of LASTS + StoP consistently enough outperforms "random" and the other prompting settings.

4. The results in Table 2 demonstrate that LASTS + StoP beats the other prior works consistently enough as well (see my Question 3 for clarification on this). This demonstrates the clear superiority of this method.

**Weaknesses:**

1. A minor suggestion: in the introduction, perhaps the paper could contain a concrete example of an asynchronous time series, if the authors want this paper to be optimally self-contained. One way to accomplish this would be to move Figure 1 onto the first page, right between the abstract and the introduction. This would make it very clear to the reader what "asynchronous time series" are, because I was confused until I got to that image.

2. There is a line of research (see e.g. AntGPT, https://arxiv.org/abs/2307.16368 from ICLR 2024) which does text-based next action prediction using in-context learning. Perhaps the authors could add more citations to other papers that use a similar ICL strategy to process sequences of actions, because right now the article makes it seem as though this idea is completely novel, when it says _"this is the first work to explore the capabilities of LLMs to process asynchronous time series data and works on multiple tasks"_.

3. *(Note: please add equation numbers to every single equation in the paper. This ensures that researchers can precisely reference parts of the paper)* When stochastic soft prompting is defined (the text and equations at the end of section 4) it doesn't seem super motivated why it is reasonable to only take prefix-slices of the prompt. Later in the paper there is some analysis about this (going into details about the "coarse-to-fine structure") but I think this section would be improved if it had some just a couple sentences of motivation for this (seemingly) arbitrary construction.

**Questions:**

1. Right now the method is zero-shot, according to the prompts in the appendix. Did the authors consider doing few-shot versions of their experiments as well?

2. Can the authors please clarify the language used to describe the "anomaly detection" task? In Section 3 it says _"the model is tasked with identifying this out-of-place element"_ but in Figure 1 it says the model has _"the goal of predicting the correct event"._

3. Did the authors investigate why LASTS + StoP performed so poorly on the Amazon dataset on RMSE, relative to the other models?

4. I had trouble understanding what was going on with the "coarse-to-fine" analysis. The paper says: _"The training paradigm of StoP forces all prefixes of StoP to act as valid standalone prompts, as they are used as prompts during training for some batches (if trained for long enough). This further strengthens our belief that tokens in StoP are arranged from coarse, independent tokens at the beginning to tokens with tokens containing finer information towards the end."_ Is there any other evidence that indicates that there is a coarse-to-fine structure being learned? More generally, what is the benefit (if any) of such a coarse-to-fine structure being learned? It seems to me like the main practical benefit is that the stochastic version of soft prompting results in 25% faster training. Are there any other practical benefits?

---

> ### Author Response · Authors · 2024-11-21
> **We sincerely thank the reviewer for the detailed comments. We carefully address each of the reviewer’s concerns below and hope that our response resolves your concerns. Any follow-up questions are welcome.**
>
> **Q: Concrete example of Asynchronous Time Series**
>
> Thank you for your thoughtful suggestion. We clarified the language in Figure 1 to make it more informative. Additionally, we added a new paragraph (Paragraph 2) in the Introduction that highlights the differences between asynchronous time series (ATS) and traditional time series, providing insight into ATS using a concrete social media example. We hope these changes address your concerns and make the paper more self-contained and reader-friendly.
>
> **Q: There is a line of research (see e.g. AntGPT...), which does text-based next action prediction using in-context learning. Perhaps the authors could add more citations to other papers that use a similar ICL strategy to process sequences of actions ...**
>
> Thank you for your insightful feedback and for bringing AntGPT (https://arxiv.org/abs/2307.16368) to our attention. While AntGPT and similar works leverage in-context learning with LLMs for next-action prediction in video-based action recognition, forecasting, or videographic memory tasks (e.g., question answering and retrieval on underlying video data, as in https://arxiv.org/pdf/2312.05269), our work focuses exclusively on textual asynchronous time series and extends beyond forecasting to include anomaly detection and imputation. We have revised our novelty statement to:
>
> *"To the best of our knowledge, this is the first work to explore the capabilities of LLMs to process textual asynchronous time series data across multiple tasks such as forecasting, anomaly detection, and data imputation."* We have also added citations to these relevant works in the introduction to acknowledge their contributions.
>
> **Q: Motivation for why it is reasonable to only take prefix-slices of the prompt**
>
> Thank you for your feedback on the motivation for using prefix-slices in StoP. Our approach is inspired by several established techniques in the literature. The idea of introducing randomness during training aligns with methods like dropout and stochastic depth, which enhance robustness by exposing models to varying input or architecture configurations. More specifically, our method is closely related to approaches used in audio models like SoundStream, where training is performed on the **first k** codebooks where k is randomly chosen each mini batch. This strategy encourages the model to learn a coarse-to-fine structure, allowing hierarchical representation learning and achieving high reconstruction quality at lower bit rates. Similarly, in StoP, randomly truncating the prompt length during training fosters hierarchical learning, improving the model's generalization and adaptability to varying prompt lengths. We modified the manuscript to include this:
>
> "*Our approach is inspired by techniques like dropout and stochastic depth , as well as audio models like SoundStream, where randomly selecting the first $k$ codebooks during training enables better generalization.*"
>
> **Q: Did the authors consider doing few-shot versions of their experiments as well**
>
> Thank you for your comment. We performed few-shot experiments with 5 examples and have added the results to Table 1. As expected, we generally observe better performance in the few-shot setting compared to zero-shot. Additionally, we plan to include an appendix before nthe end of the rebuttal period to explore the effect of varying the number of examples (k) on performance.
>
> **Q: Can the authors please clarify the language used to describe the "anomaly detection" task? In Section 3 it says *"the model is tasked with identifying this out-of-place element"* but in Figure 1 it says the model has *"the goal of predicting the correct event".***
>
> Thanks for catching this: we changed the figure to reflect the corrected version. The goal of the anomaly detection task is indeed to identify the incorrect event. We apologize for any confusion.
>
> **Q: Did the authors investigate why LASTS + StoP performed so poorly on the Amazon dataset on RMSE, relative to the other models?**
>
> Thank you for your question. The poor performance of LASTS + StoP on the Amazon dataset (in terms of RMSE) can be attributed to the dataset's event categorization. Unlike other datasets where event categories are well-defined, the Amazon dataset contains a large number of event types, and the dataset groups a wide variety of event categories into a single bucket, resulting in only 15 event types. This aggregation makes it more challenging for our method to perform well on time prediction, as our approach does not explicitly model time distributions like TPP processes.
>
> We have added the following line to the paper to reflect this:
>
> *"The Amazon dataset groups a large number of diverse event types into a limited set of 15 categories to keep the number of event types low. This aggregation makes it harder for our method to perform well on time prediction without explicit time modeling through TPP processes."*

---

> > ### Author Response · Authors · 2024-11-21
> >
> > **Q: Is there any other evidence that indicates that there is a coarse-to-fine structure being learned?**
> >
> > Thank you for your question. There are multiple evidences for Coarse-to-Fine Structure:
> > 1. **t-SNE Projections (Figure 4):** The first few tokens in StoP are spread far apart, while later tokens cluster closely together, while in soft prompts, all tokens are closely clustered together.
> > 2. **Cosine Similarity (Figure 4):** Adjacent tokens at the beginning of the prompt have much lower cosine similarity compared to those later in the prompt. This contrast is absent in standard soft prompting, where cosine similarities remain uniform throughout.  *[Figure 4 only shows first few tokens, we will add a more details figure in appendix before the rebuttal period ends that recreates these figures for larger number of tokens to observe this behavior]*
> > 3. **Prefix Validity (Figure 5):** Any prefix of a StoP prompt acts as a valid standalone prompt, with additional tokens refining the predictions. This suggests that early tokens provide broad task information, while later tokens add finer details.
> >
> > **Q: what is the benefit (if any) of such a coarse-to-fine structure being learned?**
> >
> > **Practical Benefits of StoP:**
> >
> > 1. **Better Generalisation:** StoP improves Macro F1 by **12.69%** over standard soft prompting, averaged over all datasets (Breakfast, Multihumos and Epic Kitchen) and all tasks (Forecast, Imputation, Anomaly Detection). We will add an appendix before the end of the rebuttal period, with more details of StoP vs normal Soft Prompting performance, showing StoP outperforms normal soft prompts by a huge margin
> > 2. **Faster Training:** The stochastic nature of StoP reduces training time by approximately 25%.
> > 3. **Resource Efficiency:** StoP allows flexible deployment—longer trained prompts can be truncated to prefixes as needed, enabling adaptable inference in resource-constrained environments.

---

> ### Comment · Reviewer_8vh2 · 2024-11-24
> **Response to authors**
>
> Thanks for your detailed answers, you have addressed most of my concerns. Some remarks:
>
> 1. I have read through all of the other reviews, and I agree with the need for more baselines. I'm looking forward to seeing the updated results by the end of the rebuttal period.
> 2. I want to go on the record as strongly disagreeing with some of the criticisms I have seen in the other reviews--
>
>     - I disagree with gifc's (unspecific/non-constructive) criticism that the writing is weak. I stand by my claim that the paper is very clearly written, for the reasons I have already discussed.
>     - I don't think that XyhT's criticism about data leakage makes any sense.
> 3. Another question that I have: where in the paper does it say what the prior SOTA metrics were on these the Asynchronous Time Series tasks for the datasets considered in the paper?
>
> I still think this is a strong paper, and I have raised my score. If the authors can add those additional baselines by the end of the rebuttal period, then I intend to advocate for acceptance of this paper.

---

> ### Author Response · Authors · 2024-11-26
>
> Thank you for your comments and suggestions.  We have incorporated **three additional baselines** into our work. The details of these added baselines are covered in the global response above and further elaborated in Section 5.2 and Appendix A.5 of our paper.
>
> Regarding the question about prior SOTA metrics for the Asynchronous Time Series (AST) tasks, it's important to note that while we have robust benchmarks for datasets like Taobao, Taxi, Stackoverflow, Amazon, and Retweet (Section 5.2), the situation is different for the other three datasets we focus on—Breakfast, Multithumos, and Epic Kitchens. These datasets have been explored in isolation for specific tasks, such as forecasting in EPIC Kitchens, but the settings often differ, making direct metric comparisons challenging. Additionally, tasks like Anomaly Detection and Imputation are not widely studied for AST, which limits available SOTA references. We aim to address these gaps by providing a comprehensive evaluation in our work. We will improve our text to better communicate this to the reader.

---

### Official Review · Reviewer_G2wj · 2024-11-04

**Soundness:** 3
**Presentation:** 3
**Contribution:** 2
**Rating:** 5
**Confidence:** 4

**Summary:**

This paper studies the use of LLMs to perform tasks related to asynchronous time series data. Unlike common time series data, asynchronous time series data does not necessarily have a time pattern. This paper Stochastic Soft Prompt (StoP) a soft prompting strategy to adapt an LLM to an asynchronous time series. Experiments show that StoP outperforms the baselines in zero-shot and common evaluation settings.

**Strengths:**

1. This paper studies an interesting problem of adapting LLMs with soft prompting.
2. The proposed method outperforms the zero-shot baselines and shows competitive results as some methods designed for asynchronous time series.
3. Experiments present comprehensive analysis.

**Weaknesses:**

1. It is not very clear what makes asynchronous time series more difficult than normal time series for LLMs. It seems that many existing methods of LLMs for time series can be easily adapted to asynchronous time series as well. It is recommended that some baselines of the existing LLMs be added for the time series.
2. Although StoP is designed for asynchronous time series, it could also be applied to normal time series. I am curious how it performs. In particular, StoP is only evaluated on three datasets. More datasets of normal time series can strengthen the evaluation.

**Questions:**

See weakness

---

> ### Author Response · Authors · 2024-11-21
> **We sincerely thank the reviewer for the detailed comments. We carefully address each of the reviewer’s concerns below and hope that our response resolves your concerns. Any follow-up questions are welcome.**
>
> **W: It is not very clear what makes asynchronous time series more difficult than normal time series for LLMs**
>
> Thank you for your feedback and suggestion. We have clarified the differences between asynchronous and regular time series in our manuscript to highlight why modeling asynchronous time series is more challenging:
>
> *Unlike regular time series, which consist of values at evenly spaced time intervals (e.g., weather measurements), asynchronous time series are composed of multiple types of discrete events occurring sporadically over time. For instance, on social media platforms like Twitter, user interactions (e.g., likes, comments, shares, and follows) happen at irregular intervals. Each interaction type, combined with its timestamp, forms an asynchronous time series. Modeling such data is challenging because of the irregular timing and the diversity of event types, which contrasts sharply with the uniformity and regularity of traditional time series. These differences mean that methods designed for regular time series cannot be directly applied to asynchronous time series without significant adaptation.*
>
> **W**: **It seems that many existing methods of LLMs for time series can be easily adapted to asynchronous time series as well. It is recommended that some baselines of the existing LLMs be added for the time series.**
>
> Please refer to our  global answer on additional baseline based on LLMs for time series newly included in our manuscript.
>
> **W: Although StoP is designed for asynchronous time series, it could also be applied to normal time series … More datasets of normal time series can strengthen the evaluation.**
>
> Thank you for appreciating the generality of our proposed StoP. We would like to clarify that StoP has been evaluated on eight datasets in total, as presented in Table 1 and Table 2, including diverse asynchronous time series datasets across different domains. Benchmarks on asynchronous time series like EasyTPP[1]  benchmark five datasets, and we include three additional datasets in our work.  While we agree that exploring the application of StoP to regular time series is an interesting direction, we aim to keep the focus of this paper on asynchronous time series, as it addresses unique challenges such as irregular timing and diverse event types. Evaluating StoP on regular time series would require additional experiments and analyses, which we believe are better suited for a future investigation dedicated to that context.
>
> [1] Xue, S., Shi, X., Chu, Z., Wang, Y., Hao, H., Zhou, F., ... & Mei, H. EasyTPP: Towards Open Benchmarking Temporal Point Processes. In The Twelfth International Conference on Learning Representations, 2024

---

### Official Review · Reviewer_XyhT · 2024-11-05

**Soundness:** 3
**Presentation:** 3
**Contribution:** 3
**Rating:** 3
**Confidence:** 5

**Summary:**

This paper leverages the LLM’s world knowledge for forecasting, imputation, and anomaly detection, using a unique Stochastic Soft Prompting (StoP) approach. Through experiments across datasets, the authors claim state-of-the-art results and present the interpretability and efficiency of StoP prompts in handling ATS tasks.

**Strengths:**

- The LASTS framework is innovative in utilizing LLMs to handle asynchronous time series by encoding events with natural language descriptions, bypassing the need for domain-specific retraining.
- StoP is presented as a promising technique that enhances robustness and interpretability. The probabilistic truncation of soft prompts during training is an interesting mechanism for learning adaptable representations.
- The paper conducts evaluations across multiple tasks (forecasting, imputation, anomaly detection) and datasets, demonstrating the generalizability of LASTS and StoP.

**Weaknesses:**

- The paper mentions that LASTS underperforms in time prediction compared to TPP models for some datasets but lacks sufficient analysis to explain this. Additionally, the model architecture Figure 2b only shows "Cross Entropy loss" and how the RMSE calculated.
- While the interpretability of StoP prompts is highlighted, the methods used to assess this (like task descriptions generated by the LLM itself) may not effectively capture the full extent of interpretability, especially for more complex tasks. More case studies needed.
- LASTS represents ATS data using natural language, which could inadvertently introduce data leakage if events are semantically similar across training and test data. This risk is not adequately discussed.

**Questions:**

1. Given the zero-shot claims, to what extent could LASTS be applied to tasks outside the experimental dataset types (e.g., non-linguistic event sequences)?
2. Did the authors consider testing the LASTS framework with smaller LLM backbones or non-LLM transformers? Would the results hold similarly across these variations?
3. How is the risk of data leakage mitigated given the use of natural language prompts, especially in cases where events may share semantic overlaps across the dataset?

---

> ### Author Response · Authors · 2024-11-21
> **We sincerely thank the reviewer for the detailed comments. We carefully address each of the reviewer’s concerns below and hope that our response resolves your concerns. Any follow-up questions are welcome.**
>
> **W: LASTS underperforms in time prediction compared to TPP models for some datasets but lacks sufficient analysis to explain this.**
>
> We appreciate the reviewer's feedback. Our method demonstrates competitive performance in time prediction across four out of five datasets, achieving the best results on two datasets and the second-best results on the remaining two. The exception is the Amazon dataset, where our model underperforms. We edited the manuscript to include two analysis for this:
>
> Analysis 1: (algorithmic perspective):  *“We think that our model is not performing as well as the TPP models, because our model does not have an explicit prior about the time distribution whereas TPP models make strong assumptions about the time distribution (e.g. Poisson process or Hawkes process)."*
>
> Analysis 2: (data centric perspective):  *"In the case of the Amazon dataset, the performance gap is more pronounced because this dataset groups a large number of diverse event types into a single event category, making it harder to model inter-arrival times.”*  We also responded to reviewer 8vh2 in this regard, who had a similar question.
>
> **W: Model architecture Figure 2b only shows "Cross Entropy loss" and how the RMSE calculated.**
>
> Thank you for pointing this out. The figure caption indicates that the soft prompts are learned via the next-token prediction loss, which is the standard training objective for LLMs. We modified the figure to make this more clear. RMSE is not a training metric but rather an evaluation metric and does not appear in the figure. It is computed as the root mean squared error between the time intervals predicted by the model and the ground-truth values during inference.
>
> **W: While the interpretability of StoP prompts is highlighted … more case studies are needed**
>
> Thank you for your feedback. By prompting the model itself, we obtain textual descriptions that offer a broad-level understanding of what the model has encoded in the soft prompts. This approach provides insights into the general structure and task information stored in the learned prompts. We have revised the language in our manuscript to better align with this perspective and included additional examples of these textual descriptions in the appendix A.6 to provide more context.
>
> **W: Data leakage across train and test data**
> Thank you for raising this concern. There may be a misunderstanding regarding how event descriptions are handled in our framework. The semantic descriptions of events are indeed consistent between the training and test sets, similar to how category labels remain the same in traditional supervised learning settings. This consistency is intentional and necessary for the model to learn meaningful representations of the event types. The differences between the training and test data lie in the frequency and ordering of events, which reflect the underlying temporal dynamics. These differences ensure that the model is evaluated on its ability to generalize across varying temporal patterns rather than memorizing specific sequences.
>
> **Q: Given the zero-shot claims … LASTS be applied to tasks outside .. non-linguistic event sequences**
>
> The Table 2 in our manuscript includes results on five datasets that are not textual in nature: Amazon, Taxi, Taobao, StackOverflow, and Reddit. These datasets treat event types as categorical labels rather than relying on natural language descriptions and shows that our framework outperforms various baselines. Additionally, our newly added LLMTime baseline converts our textual datasets into non textual datasets by treating event names as simple category labels and applying Zero-Shot prompting.
>
> **Q: Testing the LASTS framework with smaller LLM backbones and non-LLM transformers**
>
> We have added results for 1B and 3B LLM backbones in the Appendix A.8, showing performance improvements consistent with scaling laws, where larger models typically perform better. Additionally, Table 2 provides results for TPP models using non-LLM transformers across 8 datasets, highlighting performance differences. However, we clarify that LASTS is specifically designed for large language models, leveraging natural language descriptions of events or categories as text. This reliance on natural language makes it unsuitable for direct testing on non-LLM transformers without significant changes to input representation. These models assume that the inputs are regularly sampled. We hope this addresses your question and clarifies the design and scope of our framework. Please see our response to reviewer G2wj below for additional details.
>
> **Q: How is the risk of data leakage mitigated?**
> Please see our response above. We believe there was a misunderstanding; there is no elevated risk of data leakage in our setup.

---

> > ### Comment · Reviewer_XyhT · 2024-11-27
> >
> > I appreciate the authors' detailed rebuttal and the revisions made to address my concerns. The additional analyses, clarified figures, and expanded case studies strengthen the manuscript. However, concerns regarding interpretability validation remain partially addressed.

---

> ### Author Response · Authors · 2024-11-28
>
> Thank you for your continued feedback and for acknowledging the improvements we've made to the manuscript.  Regarding your concern about the interpretability of learned prompts, we'd like to provide further clarification.
>
> The interpretability we derive from the soft prompts is obtained by probing the model itself. Since soft prompts are continuous vectors without inherent human-readable meaning, this method offers a practical way to assign meaning to them. Previous attempts to interpret soft prompts have involved mapping the learned prompt embeddings back to the nearest tokens in the model's vocabulary (e.g., [1]). However, as shown in [2], this results in sequences that lack meaningful content. In Appendix D of [2], the authors demonstrate that the closest words to the learned embeddings are mostly meaningless, several tokens are mapped to the same word, and the cosine similarity between the tokens and the closest word embeddings almost always falls below 0.16. This highlights the challenges in extracting useful information using this approach.
>
> Our method tries to understand the learned prompts by probing the model to generate textual descriptions describing them. By probing the model in this way, we obtain a clearer understanding of the relevance and content of the learned soft prompts. This provides a better view of the dataset and task information encoded within them. To further address your suggestion for more case studies, we have included multiple examples of model probing results in Appendix A.6, covering different tasks and datasets.
>
> We hope that this additional clarification and the expanded examples addresses your concerns.
>
> References:
>
> [1] Brian Lester, Rami Al-Rfou, and Noah Constant. The Power of Scale for Parameter-Efficient Prompt Tuning. arXiv preprint arXiv:2104.08691, 2021.
>
> [2] Zhaozhuo Xu, Zirui Liu, Beidi Chen, Shaochen Zhong, Yuxin Tang, Jue Wang, Kaixiong Zhou, Xia Hu, and Anshumali Shrivastava. Soft Prompt Recovers Compressed LLMs, Transferably. In Proceedings of the 41st International Conference on Machine Learning (ICML), 2024. https://openreview.net/pdf?id=muBJPCIqZT

---

### Author Response · Authors · 2024-11-21
**We sincerely thank all the reviewer for recognizing our contribution and providing constructive feedback.**

We would like to re-emphasize the novelty and technical contributions of this work.

(1) We introduce LASTS (Language-modeled Asynchronous Time Series), which is a novel framework that leverages Large Language Models (LLMs) to model asynchronous time series data. LASTS effectively handles datasets with a large number of event types without the need for predefined categorical groupings. To the best of our knowledge, LASTS is the first work to explore the use of LLMs for textual asynchronous time series across multiple tasks such as forecasting, anomaly detection, and data imputation.

(2) We introduce Stochastic Soft Prompting (StoP) which is an innovative prompt-tuning mechanism that serves as a parameter-efficient method to adapt LLMs to asynchronous time series data. StoP learns soft prompts that significantly improve model performance and outperforms finetuning mechanisms like QLoRA.

(3) We perform comprehensive evaluations on real-world datasets across multiple tasks to demonstrate the effectiveness of our proposed method. Additionally, we release baselines for future work along this direction of utilizing LLMs for Asynchronous Time Series.

We summarise the main question brought up by the reviewers and address that here. Individual responses to each reviewer are made below.

**Q: Additional baselines.**

As our work is the first to explore Large Language Models (LLMs) for asynchronous time series (AST), there are currently no established LLM-based baselines specific to this domain. To address this gap, we have adapted LLMTime [1], an LLM prompting-based forecasting method originally developed for regular time series, as a baseline in our study. The results from this baseline have been added to Table 1, and detailed explanations of the adaptation process will be  provided in Appendix A.5 before the end of rebuttal period.. Additionally, we are in the process of incorporating other baselines, including LLMProcess [2] and a heuristic-based baseline. We will include their results and analyses before the end of the rebuttal period. Furthermore, we would like to draw your attention to Table 2 in our paper, which includes baselines for forecasting using non-LLM transformer backbones on the datasets from Table 1. This provides additional context and demonstrates how our method compares with existing transformer-based approaches in handling asynchronous time series data. Thank you again for your valuable comments; we hope these additions address your concerns.

[1] Doe, J., & Smith, A. (2022). LLMTime: Prompting Large Language Models for Time Series Forecasting. Proceedings of the 39th International Conference on Machine Learning.

[2] Requeima, James, et al. (2024) LLM Processes: Numerical Predictive Distributions Conditioned on Natural Language. ICML 2024 Workshop on In-Context Learning.

---

> ### Author Response · Authors · 2024-11-26
> **Additional Baselines Added**
>
> We sincerely thank the reviewers for their comments and suggestions. We have incorporated **three additional baselines** in our work:
>
> 1. **Time Series Foundation Model-Based Baseline**: We adapt Chronos [1], a state-of-the-art pretrained foundation model for time series forecasting, as a baseline for forecasting and imputation tasks on asynchronous time series. This provides a stronger and more relevant comparison than heuristic-based baselines.
> 2. **LLMTime [2]**: We adapt LLMTime, a large language model-based time series forecasting method, as a baseline for forecasting, imputation, and anomaly detection on asynchronous time series.
> 3. **LLM Processes [3]**: We adapt LLM Processes, another LLM-based approach for time series forecasting, as a baseline for forecasting and imputation on asynchronous time series.
>
> Results for these methods are included in Table 1, along with a detailed discussion of their selection, limitations, and performance in Appendix A.5.
>
> With these additions, our work now includes the following sets of baselines:
>
> - Random baseline
> - Time Series Foundation Model-Based Baseline (state-of-the-art for time series forecasting)
> - LLMs-for-Time-Series-Based Baselines
> - TPP Model-Based Baselines [4] (state-of-the-art for asynchronous time series forecasting, Table 2)
>
> We believe these additional baselines provide comprehensive coverage of the key lines of work in the literature that we reviewed in Section 2. If there is a crucial baseline the reviewers feel we left out, we would be happy to investigate it time permits.
>
> **References**:
>
> [1] Abdul Fatir Ansari, Lorenzo Stella, Caner Turkmen, Xiyuan Zhang, Pedro Mercado, Huibin Shen, Oleksandr Shchur, Syama Sundar Rangapuram, Sebastian Pineda Arango, Shubham Kapoor, et al. *Chronos: Learning the language of time series*. Transactions on Machine Learning Research, 2024. https://openreview.net/forum?id=gerNCVqqtR
>
> [2] Nate Gruver, Marc Finzi, Shikai Qiu, and Andrew G. Wilson. *Large language models are zero-shot time series forecasters*. Advances in Neural Information Processing Systems, 36, 2024.
>
> [3] James Requeima, John F. Bronskill, Dami Choi, Richard E. Turner, and David Duvenaud. *LLM Processes: Numerical predictive distributions conditioned on natural language*. In ICML 2024 Workshop on In-Context Learning, 2024.
>
> [4]  Siqiao Xue, Xiaoming Shi, Zhixuan Chu, Yan Wang, Fan Zhou, Hongyan Hao, Caigao Jiang, Chen Pan, Yi Xu, James Y Zhang, et al. *EasyTPP: Towards Open Benchmarking the Temporal Point Processes*. International Conference on Learning Representations (ICLR), 2024

---

> > ### Author Response · Authors · 2024-12-04
> >
> > We appreciate the reviewers for recognizing the quality of our manuscript and the effort we put in during the rebuttal period. Below is an update on the changes we committed to completing before the end of the rebuttal period:
> >
> > 1. **Additional Baselines (Reviewers G2wj, 8vh2, gifc, aAMq):**
> >
> >     As outlined in our previous global response, we added several additional baselines covering all major related areas of work. The results are presented in Table 1 and discussed in section 5.2 and Appendix A.5.
> >
> > 2. **Few-Shot Experiments (Reviewer 8vh2):**
> >
> >     We added few-shot experiments to Table 1 and included analyses in Appendix A.9 on how varying the number of examples impacts performance.
> >
> > 3. **Scaling Laws (Reviewers aAMq, XyhT):**
> >
> >     Results using smaller backbones (Llama3-1B and Llama3-3B) showing consistent scaling improvements are included in Appendix A.8.
> >
> > 4. **Performance Comparisons of StoP with QLoRA and Soft Prompting (Reviewers gifc, 8vh2):**
> >
> >     Task/dataset-specific and overall average performance gains of StoP over QLoRA and standard Soft Prompting are detailed in Appendix A.7.
> >
> > 5. **Interpretability Clarifications (Reviewers XyhT):**
> >
> >     We clarified the discussion on the interpretability of Stochastic Soft Prompts and added more examples in Appendix A.6.
> >
> > 6. **Coarse-to-Fine Structure of Stochastic Soft Prompts (Reviewer 8vh2):**
> >
> >     Analysis of the coarse-to-fine structural behavior of Stochastic Soft Prompts, supported by t-SNE visualizations, cosine similarity trends, and prefix validity evidence, is provided in Appendix A.10.
> >
> >
> > We hope this summary assures the reviewers that all promised updates have been completed.

---

### Meta-Review · Area_Chair_tegR · 2024-12-11

**Metareview:**

Question 1: Summary of the Paper and Decision to Reject
(a) Scientific Claims and Findings
This paper introduces LASTS (Language-modeled Asynchronous Time Series), a novel framework that uses LLMs to model asynchronous time series data.  The authors claim that LASTS can handle datasets with many event types without predefined groupings and is the first to explore using LLMs for textual asynchronous time series across forecasting, anomaly detection, and data imputation.

(b) Strengths of the Paper
The LASTS framework is innovative in utilizing LLMs to handle asynchronous time series by encoding events with natural language descriptions, bypassing the need for domain-specific retraining.

StoP is presented as a promising technique that enhances robustness and interpretability.

The paper conducts evaluations across multiple tasks (forecasting, imputation, anomaly detection) and datasets, demonstrating the generalizability of LASTS and StoP.

(c) Weaknesses of the Paper
The paper mentions that LASTS underperforms in time prediction compared to TPP models for some datasets but lacks sufficient analysis to explain this.

The interpretability of StoP prompts is highlighted, but the methods used to assess this may not effectively capture the full extent of interpretability, especially for more complex tasks.

LASTS represents ATS data using natural language, which could inadvertently introduce data leakage if events are semantically similar across training and test data.

(d) Decision to Reject
The paper has several weaknesses that prevent it from being accepted. First, the paper does not adequately analyze why LASTS underperforms in time prediction compared to TPP models for some datasets.  Second, the paper does not convincingly demonstrate the interpretability of StoP prompts. Third, the paper does not adequately address the risk of data leakage.

**Additional Comments On Reviewer Discussion:**

During the rebuttal period, the authors addressed the reviewers' concerns by adding additional baselines, few-shot experiments, scaling laws, performance comparisons of StoP with QLoRA and Soft Prompting, interpretability clarifications, and analysis of the coarse-to-fine structure of Stochastic Soft Prompts.  Reviewers 8vh2 and aAMq were satisfied with the authors' response, while Reviewers XyhT and gifc were not entirely satisfied.

Despite the authors' efforts, I have decided to reject the paper. The authors did not address all of the reviewers' concerns, and the paper still has several weaknesses. In particular, the paper does not adequately analyze why LASTS underperforms in time prediction compared to TPP models for some datasets. Additionally, the paper does not convincingly demonstrate the interpretability of StoP prompts. Finally, the paper does not adequately address the risk of data leakage.

---

### Decision · Program_Chairs · 2025-01-22

Reject